# EXPLORING EXTREME PARAMETER COMPRESSION FOR PRE-TRAINED LANGUAGE MODELS

**Benyou Wang**[*]
University of Padua
wang@dei.unipd.it

**Yuxin Ren**[*]
Tsinghua University
ryx20@mails.tsinghua.edu.cn

**Lifeng Shang**[†]**, Xin Jiang, Qun Liu**
Huawei Noah's Ark Lab
Shang.Lifeng,Jiang.Xin,qun.liu@huawei.com

## ABSTRACT

Recent work explored the potential of large-scale Transformer-based pre-trained models, especially Pre-trained Language Models (PLMs) in natural language processing. This raises many concerns from various perspectives, e.g., financial costs and carbon emissions. Compressing PLMs like BERT with negligible performance loss for faster inference and cheaper deployment has attracted much attention. In this work, we aim to explore larger compression ratios for PLMs, among which tensor decomposition is a potential but under-investigated one. Two decomposition and reconstruction protocols are further proposed to improve the effectiveness and efficiency during compression. Our compressed BERT [1] with 1/7 parameters in Transformer layers performs on-par with, sometimes slightly better than the original BERT in GLUE benchmark. A tiny version achieves 96.7% performance of BERT-base with 1/48 encoder parameters (i.e., **less than 2M parameters** excluding the embedding layer) and 2.7× faster on inference. To show that the proposed method is orthogonal to existing compression methods like knowledge distillation, we also explore the benefit of the proposed method on a distilled BERT.

## 1 INTRODUCTION

Pre-trained Language Models such as BERT (Devlin et al., 2018) and ALBERT (Lan et al., 2019) have significantly improved various NLP tasks with significant improvement. Much recent work (Brown et al., 2020; Narayanan et al., 2021; Fedus et al., 2021) explores the potential of super large-scale PLMs. However, such large-scale PLMs are both economically and ecologically unfriendly (Bender et al., 2021; Patterson et al., 2021). Furthermore, deployment of large-scale PLMs is also challenging since (1) a model cannot be fully deployed or stored in a single GPU server, model parallelism would consume extra time for network communication among many servers; (2) edge devices may not have enough space for storing models; (3) the long inference time cannot support real-time feedback.

Scaling down a model with negligible performance drop would facilitate the real-world applications of PLMs in a smaller size, faster inference time, and less network communication cost. For example, recent work explores quantization (Zhang et al., 2020; Bai et al., 2020), weights pruning (Hou et al., 2020), and knowledge distillation (Jiao et al., 2020; Sanh et al., 2020) for BERT (one of the most popular PLMs). We argue that existing methods cannot largely compress large-scale PLMs as stated in Sec. 2. In this paper, we aim to explore extreme parameter compression (i.e., bigger compression ratios) although they are by definition challenging.

The parameter redundancy in PLMs was demonstrated by (Kovaleva et al., 2019; Michel et al., 2019; Voita et al., 2019; Cordonnier et al., 2021), for which we divide into two groups: *intra-matrix redundancy* and *inter-matrix redundancy*. The former happens in different heads that are

---

[*]Benyou and Yuxin contributed to this work equally.

[†]Lifeng is the corresponding author.

[1]https://github.com/twinkle0331/Xcompression

calculated separately, e.g., attentions *among heads* act on a similar subspace and are therefore low-rank (Cordonnier et al., 2021) – we relate this phenomenon to the so-called 'decomposability' defined in this paper. Like self-attention layers, decomposability also holds in FFN layers – each FFN layer could be decomposed to many independent sub-FFNs (as explained in Appendix B). One example of *inter-matrix redundancy* happens across *different layers*, e.g., attention maps among layers might be similar (Clark et al., 2019; Vig, 2019; Rogers et al., 2020).

Exploration of main weight matrices in Transformer layers finds that these weight matrices are possible to be approximated in a low-rank manner – evidencing the possible *intra-matrix redundancy* and *inter-matrix redundancy*. We comprehensively analyze and compare different decomposition methods for parameter compression including matrix decomposition (denoted as II), tensor train decomposition (Oseledets, 2011) (denoted as III) and Tucker decomposition (De Lathauwer et al., 2000) (denoted as IV). The fundamental difference between them is as below. II conducts matrix factorization (e.g., SVD) for each weight matrix thanks to *intra-matrix redundancy*. Regarding *inter-matrix redundancy*, III shares the head and tail matrices while keeping the core matrix individual; IV introduces 'matrix bank' to make parameter scale being nearly constant w.r.t. the number of layers. It is concluded that Tucker decomposition (IV) is more parameter-efficient than others in terms of compression ratios. ALBERT (Lan et al., 2019) and III can be considered as *special cases* of IV.

The practical challenges of matrix/tensor decomposition for compression are twofold. First, the decomposition may result in a discrepancy between the raw weights and approximated weights, and exact decomposition is impossible with large compression ratios. Instead, Knowledge Distillation (KD) is used on the compressed model to simulate the predictions of the raw model in a loss-aware manner. Second, reconstruction may lead to additional computation costs. *An efficient reconstruction protocol* is implemented by reordering multiplication operations that also preserve the same results.

The **contributions** of this work are (1) we propose a formal framework with standardized terminology to comprehensively discuss matrix/tensor decomposition methods to compress Transformer-based language models; (2) we adopt tensor decomposition for compressing PLMs which is also faster, while existing work (Ma et al., 2019; Liu et al., 2021) did not show the potential for speedup in PLMs; (3) our compressed BERT with $1/7$ parameters in Transformer layers performs on-par with the original BERT in GLUE benchmark. Also, a tiny version achieves 96.7% performance of BERT-base with only $1/48$ parameters in Transformer layers and 2.7× faster on inference. We directly use the proposed methods on TinyBERT (Jiao et al., 2020) that is purely based on KD, since our work is **complementary** to existing compression methods like KD.

## 2 RELATED WORK

**Compressing PLMs** Although various work was proposed to design a new efficient Transformer (Tay et al., 2020), e.g., (Ma et al., 2019; Choromanski et al., 2020; Wang et al., 2020; Kitaev et al., 2020; Zaheer et al., 2020; Cao et al., 2020), in this paper, we are focusing on the compression of Transformer-based pre-trained language models. The difference is that the latter expects to reuse well-trained models, e.g., BERT and GPT (and even GPT3 (Radford et al., 2019) and PanGu-$\alpha$ (Zeng et al., 2021)) with manageable computing resources, which typically does not change the original Transformer architecture. Taking BERT, one of the most commonly-used pre-trained language models, as an example. Existing work explored quantization (Zhang et al., 2020; Bai et al., 2020), weights pruning (Lagunas et al., 2021), knowledge distillation (Jiao et al., 2020; Sanh et al., 2020), progressive module replacing (Xu et al., 2020), neural architecture search (Xu et al., 2021; Yin et al., 2021) and matrix decomposition (Noach & Goldberg, 2020).

We argue that existing compression methods (see Tab. 1) may be inadequate for extreme parameter compression, which is under-investigated. The reasons are manifold, first, the knowledge distillation-based method generally learns a new student model from scratch, which cannot inherit too much knowledge from the teacher model before distillation. Second, some methods have upper bounds of compression ratio. For example, layer-sharing ALBERT (Lan et al., 2019) shares parameters in $L$ layers with maximum $L$ times compression. Quantization replaces existing 32-bit parameters with binary parameters with a maximum of 32 times reduction. Moreover, quantization needs further hardware support, which is usually ad hoc to specific platforms. Weight pruning arguably cannot achieve a big compression ratio (McCarley et al., 2019; Michel et al., 2019).

Table 1: The comparison between compression methods for BERT. $L$ is the number of layers and $D$ is the dimension of hidden states. In the 'Compressing ratio' column in Quantization, '32' refers to that one usually uses 32-bit floating precision for a real number.

| Methods | Examples | Hardware support | Compressing ratio |
|---|---|---|---|
| Knowledge distillation | (Sun et al., 2019; Jiao et al., 2020; Sanh et al., 2020) | ✗ | $\mathbb{O}(\frac{LD}{ld})$ |
| Parameter sharing | (Lan et al., 2019) | ✗ | $L$ as upper bound |
| Quantization | (Zhang et al., 2020; Bai et al., 2020) | ✓ | 32 as upper bound |
| Weight pruning | (Hou et al., 2020) | ✗ | - |
| Matrix decomposition | (Lan et al., 2019; Noach & Goldberg, 2020) | ✗ | $\mathbb{O}(\frac{D}{d})$ |
| Tensor (Tucker/TT) decomposition | - | ✗ | $\mathbb{O}(\frac{LD^2}{ld^2})$ |

**Matrix/tensor decomposition for compression** Tensor/matrix decomposition aims to approximate a given tensor using a set of smaller tensors/matrices. It has been investigated to compress and speed up CNNs, RNNs, and Transformers for many years (Lebedev et al., 2014; Yang et al., 2017; Ye et al., 2018; Denton et al., 2014; Winata et al., 2019). Matrix decomposition (e.g., ALBERT (Lan et al., 2019) in the embedding layer and (Noach & Goldberg, 2020)) could decrease parameter scale with a linear factor depending on the selected rank. More advanced tensor decomposition approaches can be implemented by tensor network, which has recently been used to compress general neural networks (Gao et al., 2020; Novikov et al., 2015), compress embedding layer (Khrulkov et al., 2019; Hrinchuk et al., 2020; Panahi et al., 2019).

Recently, Ma et al. (2019) redesigned a new 'Self-Attention Network' (SAN) in Transformer architecture inspired by block-term tensor decomposition. The compression ratio is limited since the majority of parameters in Transformer comes from another module called 'Feed-Forward Network' (FFN) instead of SAN; moreover, the model does not have the potential for speedup regardless of compression ratios since the time complexity is closed to vanilla Transformer. Noach & Goldberg (2020) reparameterized each weight matrix using matrix decomposition and further distill the compressed models, which nearly achieves a compression ratio of 1.7. Liu et al. (2021) adopted matrix product operators to reparameterize each group of weight matrices in embedding, SAN, and FFN, and only a small part of tensors of MPO (called 'auxiliary tensors') are fine-tuned in downstream tasks. The compression ratio of the total parameter is negligible and the inference might be slow. Those works inspire us to explore in depth extreme parameter compression for large-scale PLMs.

We argue that compression ratio of existing work using matrix/tensor decomposition (Ma et al., 2019; Liu et al., 2021; Noach & Goldberg, 2020) for PLMs is relatively-small; most of them do not have speedup effect, limiting their applications in large-scale PLMs. The potential to compress PLMs with matrix/tensor decomposition is under-investigated. In this work, we adopt tensor decomposition, to cubically compress the parameters of PLMs.

## 3 MOTIVATIONS FOR PARAMETER COMPRESSION

Pre-trained language models are typically a stack of multiple Transformer (Vaswani et al., 2017) layers that consist of a Self-Attention Network (SAN) module and a Feed-Forward Network (FFN) module, see App. A for Transformer. Sec. 3.1 will introduce an important property called 'decomposability' for SAN and FFN, which indicates each sub-component in SAN or FFN is independently calculated without interactions between sub-components and they may therefore be redundant.

### 3.1 DECOMPOSABILITY IN TRANSFORMER

A computing module $f$ is **decomposable** if its sub-components $\{g_1, g_2, \cdots g_H\}$ could be independently calculated without interactions: $f(x) = \delta\big(g_1(x), g_2(x), \cdots, g_H(x)\big)$. Usually, $\delta$ is a simple operation that has negligible computing cost compared to $\{g_h\}$. Especially, backpropagation between sub-components is independent if $\delta$ is concatenation or addition. Sub-components in $f$ could be calculated in parallel without interactions. We will examine if SAN and FFN are *decomposable*.

**Decomposability in SAN** Following (Hou et al., 2020), SAN could be decomposed as a sum of the output of every head. For the query/key/value/output transformations parameterized by

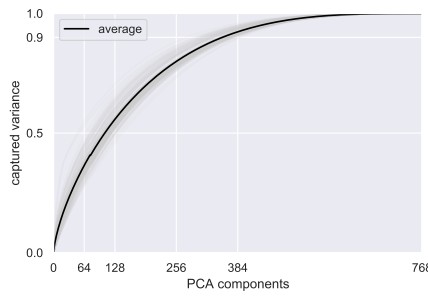

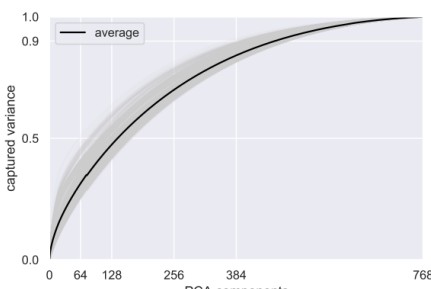

(a) PCA for each single weight matrix  (b) PCA for a pair of matrices along columns

Figure 1: PCA for existing weight block matrices in BERT-base. We got nearly similar results in Fig. 5 for paired matrices along rows and columns, as shown in App. C.

$W^Q/W^K/W^V/W^O$, we divide them as $N_h$ heads: $\{W_h^Q\}^{N_h}, \{W_h^K\}^{N_h}, \{W_h^V\}^{N_h}, \{W_h^O\}^{N_h}$. A single head is calculated as

$$\text{Att}_h(\mathbf{X}) = \text{Softmax}(\frac{1}{\sqrt{d}}\mathbf{X}W_h^Q W_h^{K^T}\mathbf{X}^T)\mathbf{X}W_h^V W_h^{O^T} \tag{1}$$

Then $\text{SAN}(\mathbf{X}) = \sum_{h=1}^{N_h} \text{Att}_h(\mathbf{X})$, indicating SAN is decomposable. Cordonnier et al. (2021) argued that the attentions among heads learn redundant key/query projections due to independent calculation.

**Decomposability in FFN**  The two weight matrices in FFN, denoted as $W^{In}$ and $W^{Out}$.

$$\text{FFN}(\mathbf{X}) = \sum_{h=1}^{4D} \text{GeLU}(\mathbf{X}W_{:,h}^{In} + b_h^{In})W_{h,:}^{Out} + b^{Out} \tag{2}$$

As Geva et al. (2020) points out, FFN also operates as a key-value mechanism similar to SAN: one can consider the input of FFN as a query vector, and two linear layers of FFN as keys and values, respectively. There may exist some similar key-value pairs that may introduce redundancy.

**Remarks on decomposability** Decomposability does not necessarily make sub-components being complementary, especially without orthogonality-like constraints between them. Sub-components may learn similar patterns, leading to some redundancy (Michel et al., 2019). For example, Hou et al. (2020) demonstrated that BERT could be pruned in both width and height without performance drop.

Moreover, BERT is over-parameterized in the sense that the number of training examples in downstream tasks is much fewer (393k training examples for the biggest task in GLUE, i.e., MNLI) comparing to the enormous number of parameters (110M parameters for BERT-base) – this was thought problematic according to the traditional machine learning theory (Vapnik, 2013). We believe that the pre-trained language models in downstream tasks could be compressed.

### 3.2 EXPLORATION OF AN PRE-TRAINED TRANSFORMER

Beyond the decomposability of Transformer, we conduct exploration on a widely-used Transformer-based PLM, i.e., BERT (Devlin et al., 2018). Technically, we calculate the captured variance ratio by Principal Component Analysis (PCA), as an indicator to measure the parameter redundancy.

The main weight matrices of Transformer layers are $\{W^Q, W^K, W^V, W^O, W^{In}, W^{Out}\}$ [2]. Sec. 2 and Appendix B show that FFN could be separately calculated in a *multi-head fashion*. We could sequentially split both $\{W^{In}\}$ and $\{W^{Out}\}$ into four groups like $\{W_h^{In}\}^{h=4}$ and $\{W_h^{Out}\}^{h=4}$

---

[2]We exclude embedding layer for compression, as (Ben Noach & Goldberg, 2020) did. Note that the lookup operation in embedding layers is fast; therefore, decomposing embedding will be more time-consuming since it involves additional computationally-expensive matrix multiplication. Moreover, this paper focuses on the core components in Transformer, namely SAN and FFN, which are the majority parameters that also increase linearly with network depth; while the parameter scale in embedding layer is constant w.r.t. network depth.

Figure 2: The three methods for parameter compression. To compress the raw weights $\mathbf{W}^{\mathrm{I}}$, II decomposes each matrix in $\mathbf{W}^{\mathrm{I}}$ into small matrices, i.e., two 'narrow' matrices and a small square matrix. III further shares the two 'narrow' matrices for all weights. IV introduces a matrix bank for these small square matrices, making parameter scale nearly constant w.r.t. the number of layers.

respectively. By doing so, we could reconcile all weight matrices to be of same shape (i.e. $D \times D$). Here, we could get **12** $D \times D$ weight blocks for each Transformer layer, **4** for SAN, and **8** for FFN.

In Fig. 1 we could find the **intra-matrix and inter-matrix redundancy**: Figure 1a shows that half dimensions could capture more than 90% variance of all weight matrices, this confirms our statement in Sec. 3.1. Furthermore, we also study the redundancy between two matrices by conducting PCA on the concatenated matrix between two arbitrarily paired weight matrices. See Fig. 1b, half dimensions could capture nearly 80% variance, which suggests some possibility to compress **inter-matrix redundancy**. This inter-matrix redundancy may be twofold: (1) subFFNs are decomposable; (2) calculations (e.g., attention maps) in different layers may be similar. Regarding the latter, some existing works like RNNs (Goyal & Bengio, 2020), Neural Ordinary Differential Equations (ODE) (Chen et al., 2018), and cross-layer sharing ALBERT (Lan et al., 2019) show that it could work even with the parameter equivalence hypothesis among layers.

## 4 A GENERAL FRAMEWORK FOR PARAMETER COMPRESSION

The observation that the main weight blocks in BERT could be approximated in a low-rank manner (thanks to the intra-matrix and inter-matrix redundancy) inspires us to use decomposition. Here we introduce and compare some standard decomposition methods (see App. D) in compressing PLMs.

### 4.1 EXPLORING PARAMETER COMPRESSION

In principle, SANs and FFNs could be separately compressed; in this work, we additionally explore stacking SAN weights and FFN weights together as a unified protocol since each weight block has an identical shape (i.e., $D \times D$). The main weight matrices in the $j$-th Transformer layer are

$$\mathbf{W}^{(j)} = \left[ \{\boldsymbol{W}^Q, \boldsymbol{W}^K, \boldsymbol{W}^V, \boldsymbol{W}^O\} \oplus \{\boldsymbol{W}_h^{In}\}^{h=4} \oplus \{\boldsymbol{W}_h^{Out}\}^{h=4} \right]_j \in \mathbb{R}^{12 \times D \times D}. \tag{3}$$

Weights of a $L$-layers Transformer are stacked as a 3-rd order tensor in $\mathbb{R}^{12L \times D \times D}$. The original non-decomposed weights is called I: $\mathbf{W}^{\mathrm{I}} = \{\mathbf{W}^{(j)}\}_{j=1}^{L} \in \mathbb{R}^{12LD^2}$. Each weight matrix in $\mathbf{W}^{\mathrm{I}}$ is $\mathbf{W}_i^{\mathrm{I}} = \mathbf{W}_i \in \mathbb{R}^{D \times D}$. Here, we explore standard decomposition methods including matrix decomposition, tensor train decomposition (Oseledets, 2011) and Tucker decomposition (De Lathauwer et al., 2000).

**II: matrix decomposition** Motivated by **intra-matrix redundancy**, one can adopt *Matrix decomposition* to factorize/approximate a matrix into some smaller ones. A typical example is singular value decomposition (SVD), called 'II-$\alpha$', for each $D \times D$ matrix $\mathbf{W}_i \in \mathbf{W}^{\mathrm{I}}$,

$$\mathbf{W}_i \approx \mathbf{W}_i^{\mathrm{II}} = \mathbf{U}_i \mathbf{\Sigma}_i \mathbf{V}_i \in \mathbb{R}^{D \times D} \tag{4}$$

One can also drop the diagonal $\mathbf{\Sigma}_i$ by decomposing it into two parts that are multiplied to $\mathbf{U}_i$ and $\mathbf{V}_i$ [3], namely $\mathbf{W}_i \approx \mathbf{U}_i \mathbf{V}_i$, denoted as 'II-$\beta$'. $\mathbf{U}_i \in \mathbb{R}^{D \times d}$ and $\mathbf{V}_i \in \mathbb{R}^{d \times D}$ and usually $d < D$. Since the compression ratio is $\frac{D^2}{2Dd}$ with reducing the rank from $D$ to $d$, the preserved rank of the approximated matrix linearly decreases with the compressing rates.

---

[3] By decomposing $\mathbf{\Sigma}_i$ into two diagonal matrices, each of which has diagonal elements that are the square root of $\mathbf{\Sigma}_i$. By multiplying these two diagonal matrices to $\mathbf{U}_i$ and $\mathbf{V}_i$. $\mathbf{W}_i \approx \mathbf{U}_i \mathbf{V}_i \in \mathbb{R}^{D \times D}$.

Table 2: Overview of methods. The most expensive term in space complexity is in bold.

| - | Methods | Space complexity | Difference | Reference |
|---|---|---|---|---|
| I | - | $\mathbf{12LD^2}$ | raw | (Devlin et al., 2018) |
| II-$\alpha$ | Matrix decomposition | $\mathbf{24LDd} + 12Ld^2$ | w/ low-rank approximation | - |
| II-$\beta$ | Matrix decomposition | $\mathbf{24LDd}$ | w/t diagonal matrices | (Ben Noach & Goldberg, 2020) |
| III | Tensor-train decomposition | $\mathbf{12Ld^2} + 2Dd$ | w/ parameter sharing | - |
| IV | Tucker decomposition | $\mathbf{ld^2} + 12Ll + 2Dd$ | w/ matrix bank | - |

**III: tensor train decomposition**    Inspired by the **inter-matrix redundancy**, one could expect to share weights among matrices. The biggest terms in Eq. 4 are $\mathbf{U}_i$ and $\mathbf{V}_i$ while $\{\mathbf{\Sigma}_i\}$ is relatively small since $\mathbf{\Sigma}_i \in \mathbb{R}^{d \times d}$ and $d$ is relatively small compared to D. We could share $\{\mathbf{U}_i\}$ and $\{\mathbf{V}_i\}$ among matrices to save parameters. This results in

$$\mathbf{W}_i \approx \mathbf{W}_i^{\mathrm{III}} = U\mathbf{\Sigma}_i V \in \mathbb{R}^{D \times D} \tag{5}$$

Here, $\{\mathbf{\Sigma}_i\}$ are not necessarily diagonal. This results in a tensor-train (TT) decomposition (Oseledets, 2011) [4]. One can also consider higher-order TT decomposition (i.e., a longer chain for tensor multiplications) which could be more parameter-efficient; this often needs to reshape the raw tensor into a higher-order tensor with heuristics. However, it is more time-consuming and costs more GPU memory during training, which we leave as future work.

**IV: Tucker decomposition**    In Eq. 5, the biggest term is the $\{\mathbf{\Sigma}_i\} \in \mathbb{R}^{12L \times d^2}$, especially the number of layers may be large in practice (e.g., $L = 24$ for BERT-large). To this end, we propose a fixed-sized *matrix bank* such that a weight matrix is considered as a linear combination of these matrices inside the bank, making the parameter scale become nearly a constant with respect to the number of layers. Namely,

$$\mathbf{W}_i \approx \mathbf{W}_i^{\mathrm{IV}} = U(P_i\mathbf{C})V \in \mathbb{R}^{D \times D} \tag{6}$$

where $\mathbf{C} \in \mathbb{R}^{l \times d^2}$ is a matrix bank with a size of $l$, each matrix is assigned with a weight vector $P_i \in \mathbb{R}^{1 \times l}$. ALBERT (Lan et al., 2019) could be considered as a special case of IV, see App. G.

## 4.2    COMPARISON BETWEEN I,II, III, AND IV

The comparison in parameter scale between these decomposition methods is in Tab. 2. Since $D > d$ and $L > l$, we could generally conclude that the parameter scales decrease from I, II, III to IV. We can observe that marginal parameter cost to add a new layer in IV is nearly $12l$, which is negligible compared to the other parameters. During the inference phase, the terms that do not involve batch size $b$ or sequence length $n$ could be calculated in an offline way only once before starting inference, which costs more storage but gets slightly acceleration – since the main purpose of this work is to compress models, we ignore it in this work but encourage doing it in speed-sensitive scenarios.

## 5    EXTREMELY COMPRESSING BERT USING TENSOR DECOMPOSITION

### 5.1    DECOMPOSITION PROTOCOL

IV reduces space complexity from $\mathcal{O}(12LD^2)$ to $\mathcal{O}(ld^2 + 12Ll + 2Dd)$ where $d < D$ and $l < 12L$. $l$ determines to which degree we want to share Transformer parameters among all modules, a flexible factor to smoothly transform vanilla BERT to layer-shared BERT (or called 'ALBERT' (Lan et al., 2019)). $d$ determines the expressive power (rank) of each linear transformation (originally $D \times D$).

The decomposition protocol does not change the raw architecture of BERT, alternatively, it introduces a new reparameterization of the existing weights. However, the approximated weights $\mathbf{W}^{\mathrm{IV}}$ usually

---

[4] A tensor-train decomposition is to approximate a high-order tensor with a product of many smaller three-order tensors – except for the first and last ones being matrices. Here, for a three-order tensor $\mathbf{W} \in \mathbb{R}^{12L \times D \times D}$, it is approximated by $\mathbf{W} \approx U\mathbf{G}V$ and shape transpose, where $U \in \mathbb{R}^{D \times r_1}$, $\mathbf{G} \in \mathbb{R}^{r_1 \times 12L \times r_2}$, and $V \in \mathbb{R}^{r_2 \times D}$. For a specific slice of $\mathbf{W}$, $\mathbf{W}_i \approx U\mathbf{G}_{:,i,:}V$. $r_1$ and $r_2$ are the 'TT ranks'.

Table 3: Computational complexity with different order of matrix multiplication

| - | Multiplication order | Computing complexity |
|---|---|---|
| IV-1 | $\mathbf{X}(\boldsymbol{U}(\boldsymbol{P}_i\mathbf{C})\boldsymbol{V})$ | $\mathcal{O}(bnD^2 + Dd^2 + D^2d + ld^2)$ |
| IV-2 | $(\mathbf{X}\boldsymbol{U})(\boldsymbol{P}_i\mathbf{C})\boldsymbol{V}$ | $\mathcal{O}(2bnDd + bnd^2 + ld^2)$ |
| IV-3 | $(\mathbf{X}\boldsymbol{U})((\boldsymbol{P}_i\mathbf{C})\boldsymbol{V})$ | $\mathcal{O}(2bnDd + Dd^2 + ld^2)$ |

are not exactly equal to the raw weights $\mathbf{W}^{\mathrm{I}}$. Moreover, the tiny decomposition discrepancy of weight matrices in low-layer may lead to an accumulated difference in the final output due to the multiple-layer neural network architecture [5]. In this work, we propose to use knowledge distillation to simulate the final output of raw models.

$$f_{\mathbf{W}^{\mathrm{I}}}(x) \approx f_{\mathbf{W}^{\mathrm{IV}}}(x) \tag{7}$$

$f_{\mathbf{W}^{\mathrm{I}}}$ is the raw BERT model and $f_{\mathbf{W}^{\mathrm{IV}}}$ is the compressed one. We argue that approximation in prediction (like knowledge distillation in Eq. 7) is more important than approximation in weights. Such a loss-aware strategy in compression could be found in quantization (Hou et al., 2018).

## 5.2 RECONSTRUCTION PROTOCOL

A slice of $D \times D$ parameter block is represented as matrix product, $\mathbf{W}_i^{\mathrm{IV}} \approx \boldsymbol{U}(\boldsymbol{P}_i\mathbf{C})\boldsymbol{V} \in \mathbb{R}^{D \times D}$. For an input $\mathbf{X} \in \mathbb{R}^{b \times n \times D}$ where $b$ is the batch size and $n$ is the sequence length, an output of linear transformation between $\mathbf{X}$ and a $D \times D$ parameter block will be $\mathbf{Y} = \mathbf{X}\mathbf{W}_{i;;} = \mathbf{X}\boldsymbol{U}(\boldsymbol{P}_i\mathbf{C})\boldsymbol{V}$. Since matrix multiplication is **associative** [6], different multiplication order will not affect the final result but their computational complexity may be different [7]. One can see the computational complexity for multiplication order in Tab. 3. In practice, the batch size $b$ will be set as big as possible to increase data throughput and make training more stable, we could conclude that IV-3 is more efficient than IV-2. IV-3 is more efficient than IV-1 when $D > 2d$; in practice, $D$ is typically much bigger than $d$ and $D > 2d$. In conclusion, setting IV-3 is most efficient in this scenario.

## 6 EXPERIMENTS

### 6.1 SETTINGS

**Decomposition** For BERT-base ($L = 12, D = 768,$ ), $\mathbf{W}^{\mathrm{I}} \in \mathbb{R}^{144D^2}$ is decomposed into a core tensor and three factor matrices (see Fig. 2), its reconstruction could be seen in Sec. 5.2.

**Knowledge distillation** As (Jiao et al., 2020; Zhang et al., 2020; Bai et al., 2020) did, we use *two-stage knowledge distillation* for the compressed model. At General Distillation (GD) stage, we adopt Knowledge Distillation (KD) for the compressed model to simulate the last-layer hidden states and last-layer attention maps of the general teacher model (BERT-base). At the second stage, we adopt Task-specific Distillation (TD) to simulate the logits of a task-specific BERT model (e.g., fine-tuned on MNLI task). In GD, compressed models are trained with two epochs. In TD, we also augment training data by randomly replacing a random word with a similar word according to either word vector similarity using Glove (Pennington et al., 2014) or the predicted logistics of BERT when masking the target word, see more details in (Jiao et al., 2020).

**GLUE evaluation** GLUE (Wang et al., 2018) (see App. I for more details) includes datasets for single document classification and sentence pair classification. Fine-tuning and evaluation on GLUE follows the settings from Huggingface (Wolf et al., 2019). The best-performed model is selected according to the *dev* set, where we select the learning rate in $[1e\text{-}5, 2e\text{-}5]$ and batch size in $[16, 32]$.

---

[5] Our experiments also show that a direct decomposition results in very low performance, see Tab. 6.

[6] For a sequence of matrices (e.g., $[\boldsymbol{A}, \boldsymbol{B}, \boldsymbol{C}]$), matrix multiplication with different calculation orders results in a identical result, i.e., $(\boldsymbol{A}\boldsymbol{B})\boldsymbol{C} = \boldsymbol{A}(\boldsymbol{B}\boldsymbol{C})$

[7] In this paper, we define the computational complexity of a matrix multiplication between a $n \times m$ matrix and a $m \times p$ matrix as $\mathcal{O}(nmp)$, corresponding to the number of performed multiplication operations.

Table 4: Experimental results on *test* set in GLUE. 'Para.' counts the parameters in encoder layers, excluding the embedding layer and prediction layer; Note the compression ratios will become smaller when considering parameters in the embedding layer. Requests Per Second (RPS) is Throughput calculated by a single Nvidia V100 GPU (16G) using full GPU memory, see App. J for actual inference time. The single numeric suffix in BERT-III is the dimension rank $d$; the two numeric suffixes in BERT-IV correspond to the layer rank $l$ and dimension rank $d$ in IV respectively. The evaluation metrics follow the official GLUE benchmark (Wang et al., 2018). The best performance of each task is **bold**. See App. L for the tailored comparison with (Ben Noach & Goldberg, 2020) since (Ben Noach & Goldberg, 2020) used nonstandard evaluation metrics in GLUE. (Lan et al., 2019) and (Mao et al., 2020) did not use all tasks in GLUE, we use ♠, ♥, and ♣ to calculate the average for their selected tasks. '[†]' means that these methods have a same architecture that has identical parameters, FLOPS, and RPS.

| Model (our models in bold) | Para. | FLOPS | RPS | SST-2 acc | MNLI acc | MRPC F1 | QNLI acc | QQP F1 | RTE acc | STS-B spear. | Avg all | ♠ | ♥ | ♣ |
|---|---|---|---|---|---|---|---|---|---|---|---|---|---|---|
| BERT-base (Devlin et al., 2018) ( BERT-I ) | 86.0M | 22.5B | 420.1 | 93.4 | 83.9/83.4 | 87.5 | 90.9 | 71.1 | 66.4 | **85.2** | 82.7 | 88.7 | 83.5 | 84.5 |
| **BERT-III** -384 | 23.0M | 22.5B | 452.2 | 93.4 | **84.6/83.7** | **88.1** | 90.5 | 71.9 | 68.1 | 83.9 | **83.2** | **89.0** | **84.0** | **84.8** |
| **BERT-III** -64 | 1.8M | 4.3B | 1143.6 | 91.9 | 80.1/79.6 | 85.5 | 87.7 | 70.7 | 63.3 | 80.7 | 80.0 | 86.0 | 81.0 | 82.0 |
| **BERT-IV** -72-384 | 12.3M | 22.5B | 452.3 | 93.1 | 83.9/83.2 | 87.5 | 90.2 | 71.6 | 67.3 | 83.6 | 82.6 | 88.5 | 83.5 | 84.4 |
| **BERT-IV** -36-256 | 3.9M | 15.2B | 596.9 | 92.7 | 82.5/81.8 | 87.1 | 88.9 | 71.4 | 65.2 | 81.8 | 81.4 | 87.6 | 82.3 | 83.5 |
| **BERT-IV** -36-128 | 1.9M | 8.0B | 863.0 | 92.4 | 81.1/80.2 | 86.5 | 88.3 | 71.9 | 64.4 | 81.4 | 80.8 | 86.8 | 81.7 | 82.8 |
| ALBERT (Lan et al., 2019) ♠ | 7.6M | 22.5B | 434.0 | 90.6 | 82.0/- | - | - | - | - | - | - | 86.4 | | |
| matrix decomposition (Noach & Goldberg, 2020) ( BERT-II) ♥ | 41.8M | 14.6B | 656.8 | 92.9 | - | - | 90.8 | - | 67.8 | - | - | - | 83.8 | - |
| matrix decomposition-1 (Mao et al., 2020) ♣ | 34.5M | - | - | 87.6 | 77.7/ 77.4 | - | 84.3 | 65.7 | - | - | - | - | - | 78.5 |
| matrix decomposition-2 (Mao et al., 2020) ♣ | 17.3M | - | - | 82.8 | 71.8/ 71.8 | - | 75.4 | 60.3 | - | - | - | - | - | 72.4 |
| TernaryBERT (Zhang et al., 2020) | 86.0M | - | - | 93.4 | 83.1/82.5 | 86.9 | 90.2 | 71.0 | **68.9** | 83.1 | 82.4 | | | |
| BinaryBERT (Bai et al., 2020) | 86.0M | - | - | 91.9 | 84.1/83.5 | 85.9 | 89.8 | 71.6 | 67.3 | 82.3 | 82.0 | | | |
| BERT- 6layer[†] | 43.5M | 11.3B | 837.2 | 90.7 | 80.4 / 79.7 | 85.9 | 86.7 | 69.2 | 63.6 | 80.0 | 79.6 | | | |
| Vanilla KD (Hinton et al., 2015)[†] | 43.5M | 11.3B | 837.2 | 91.5 | 80.2 / 79.8 | 86.2 | 88.3 | 70.1 | 64.7 | 80.3 | 80.1 | | | |
| BERT-PKD (Sun et al., 2019)[†] | 43.5M | 11.3B | 837.2 | 92.0 | 81.5 / 81.0 | 85.0 | 89.0 | 70.7 | 65.5 | 81.6 | 80.8 | | | |
| BERT-of-Theseus (Xu et al., 2020)[†] | 43.5M | 11.3B | 837.2 | 92.2 | 82.4 / 82.1 | 87.6 | 89.6 | 71.6 | 66.2 | 84.1 | 82.0 | | | |

Table 5: GLUE results on *test* set TinyBERT-IV and comparison with KD based methods.

| Model (our models in bold) | Para. | FLOPS | RPS | SST-2 acc | MNLI acc | MRPC F1 | QNLI acc | QQP F1 | RTE acc | STS-B spear. | Avg |
|---|---|---|---|---|---|---|---|---|---|---|---|
| TinyBERT-6layer | 43.5M | 11.3B | 837.2 | **93.1** | **84.6/83.2** | 87.3 | **90.4** | 71.6 | **70.0** | 83.7 | **83.0** |
| **TinyBERT-IV-72-384** | 12.3M | 11.3B | 899.9 | 92.0 | 83.1/82.2 | **87.7** | 89.1 | **71.7** | 65.3 | 81.6 | 81.6 |
| **TinyBERT-IV-72-256** | 6.2M | 7.6B | 1188.4 | 92.0 | 82.7/81.9 | 86.7 | 87.9 | 70.9 | 65.5 | 81.0 | 81.1 |

## 6.2 RESULTS

As shown in Tab. 4, our decomposed BERT with layer rank 144 and dimension rank 384, called 'BERT-III-384', outperforms the BERT-base, with only $1/7$ parameters in Transformer layers and slightly bigger throughout. BERT-IV-72-384 performs on-par with raw BERT, which is slightly worse than BERT-III-384 due to the smaller size. Observe that a bigger rank (both for layer mode and dimension mode) usually consistently leads to better performance. BERT-III-64 achieves 96.7% performance (82.7 vs. 80.0) with only $1/48$ parameters of Transformer layers and 2.7× speedup.

Tab. 4 shows BERT-IVs are smaller than existing parameter sharing method (Lan et al., 2019) and decomposition method (Noach & Goldberg, 2020; Mao et al., 2020). BERT-III -384/ BERT-IV -72-384 achieve comparable or even slightly better results than (Noach & Goldberg, 2020). BERT-IV outperforms (Mao et al., 2020) with a large margin. BERT-IV outperforms ALBERT – the latter needs training from scratch (1M training steps) while the former does not (less than 0.2M steps).

Observe that BERT-III-384 outperforms BERT-IV-72-384 since the former has more parameters and therefore is more expressive. Note that BERT-III-384 and BERT-IV-d-384 have nearly identical RPS and inference latency. Note, we take BERT-IV-36-128 as an example to compare with matrix decomposition (a.k.a, II, which is implemented by Noach & Goldberg (2020) with a rank of 245, denoted as BERT-IV-245), BERT-IV-36-128 is faster (see RPS in Table 4 and inference time in 8), smaller, and better-performed (see Table 10 for full performance comparison) than BERT-IV-245, evidencing the advantage of BERT-IV over matrix decomposition for compression.

To contextualize BERT-III/BERT-IV with other compression methods like knowledge distillation, Tab. 4 shows that BERT-III/BERT-IV achieves comparable performance with knowledge distillation methods (Sun et al., 2019; Xu et al., 2020; Jiao et al., 2020) while with fewer parameters. Our results

Table 6: Ablation experiments of knowledge distillation (KD) (including GD and TD). The test set of GLUE with the setting 72-384 is reported. The best performance of each task is **bold**.

| Setting | SST-2 acc | MNLI acc | MRPC F1 | QNLI acc | QQP F1 | RTE acc | STS-B spear. | Avg |
|---|---|---|---|---|---|---|---|---|
| GD + TD | **93.1** | **83.9/83.2** | **87.5** | **90.2** | **71.6** | **67.3** | **83.6** | **82.6** |
| GD + finetune | 90.9 | 81.7/80.8 | 83.8 | 88.8 | 69.9 | 63.6 | 80.7 | 80.0 |
| fine-training w/o KD | 49.9 | 35.6/36.5 | 79.9 | 50.5 | 55.0 | 49.7 | 11.3 | 47.3 |

Table 7: Experiment results on *test* of GLUE with SAN and FFN.

| Model | Para. | FLOPS | RPS | SST-2 acc | MNLI acc | MRPC F1 | QNLI acc | QQP F1 | RTE acc | STS-B spear. | Avg |
|---|---|---|---|---|---|---|---|---|---|---|---|
| BERT-base ( BERT-I) | 86.0M | 22.5B | 420.1 | **93.4** | 83.9/83.4 | 87.5 | **90.9** | 71.1 | 66.4 | **85.2** | 82.7 |
| BERT-IV -72-384 | **12.3M** | 22.5B | 452.3 | 93.1 | 83.9/83.2 | 87.5 | 90.2 | 71.6 | 67.3 | 83.6 | 82.6 |
| BERT-IV-FFN-48-384 | 37.1 M | 22.5B | **463.7** | 93.1 | 84.5/84.1 | **88.0** | 90.7 | **71.9** | **69.3** | 83.1 | **83.1** |
| BERT-IV-SAN-24-384 | 61.9M | 22.5B | 419.2 | 92.9 | 84.5/83.7 | 86.0 | 90.8 | 71.8 | 66.9 | 82.5 | 82.4 |

is also comparable to quantization methods Zhang et al. (2020); Bai et al. (2020) that use 2-bit or 3-bit weights, and pruning Lagunas et al. (2021) (see Tab. 11 in App.L).

To show *the proposed method is orthogonal to existing compression methods like pure knowledge distillation*, we further explore the proposed method in TinyBERT (Jiao et al., 2020), called 'TinyBERT-IV'. Tab. 5 shows performance loss to compress TinyBERT (degrading from 83.0 to 81.6 in TinyBERT-IV-72-384) is bigger than compressing raw BERT (degrading from 82.7 to 82.6 in BERT-IV -72-384). This is probably due to smaller redundancy in TinyBERT compared to BERT.

## 6.3 ANALYSIS

**Ablation on the necessity of knowledge distillation**  Tab. 6 shows that both GD and TD are essential for an effective decomposed BERT. In particular, the overall performance decreases from 82.6 to 80.0 by removing TD. Note that the model will collapse if we directly take decomposed BERT for fine-tuning without knowledge distillation.

**Decomposition on FFNs or SANs**  For FFNs and SANs, we use the half size of matrix bank (i.e., 24 for SANs and 48 for FFNs) and half dimension rank (i.e., 384) respectively. The two settings are called 'BERT-IV-FFN-48-384' and 'BERT-IV-SAN-24-384'. Tab. 7 shows that solely compressing SANs or FFNs could nearly achieve on par with the raw model since smaller compression ratios are achieved. In detail, FFNs are slightly easier to be compressed even with a big compression ratio comparing to SANs. It is intriguing to notice that BERT-IV-72-384 outperforms BERT-IV-SAN-24-384, although the former additionally compresses FFNs and has much fewer parameters. The reason may be that the size of the matrix bank in BERT-IV-72-384 (i.e., 72), which is shared between FFNs and SANs, is bigger than its counterpart in BERT-IV-SAN-24-384 (i.e., 24). This can shed some light on the benefit to stacking FFNs and SANs together, see more discussions in App F.

## 7 CONCLUSION AND FUTURE WORK

To largely compress PLMs, we also comprehensively compare many matrix/tensor decomposition methods and conclude tensor decomposition has the potential for extreme parameter compression. We therefore efficiently implemented tensor decomposition inspired compression for BERT, thanks to an efficient reconstruction protocol. To compensate for the decomposition discrepancy, knowledge distillation is used to simulate the output of raw BERT, which is demonstrated to be essential in ablation study. Our method with $1/7$ parameters could achieve comparable with the original model, with slight speedup. A tiny version achieves more than $96.7\%$ performance of BERT-base with $1/48$ parameters in Transformer layers and $2.7\times$ faster on inference. In the future, we expect compression of PLMs to shift purely encoder-based language models (e.g., BERT) to decoder language models, since the latter has been designed as big as we could afford, e.g. GPT3 (Radford et al., 2019). The potential of the proposed methods for compressing larger model is discussed in App. N. Furthermore, hybrid methods by mixing knowledge distillation, quantization, parameter sharing, weight pruning, and matrix/tensor decomposition together are potential in practice.

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

# A    BACKGROUND OF TRANSFORMER AND BERT

## A.1    TRANSFORMER

A Transformer layer (see Fig. 3) consists of a self-attention (SAN) module and a feed-forward network (FFN) module. An input $X$ for SAN will be linearly transformed into query, key, value, and

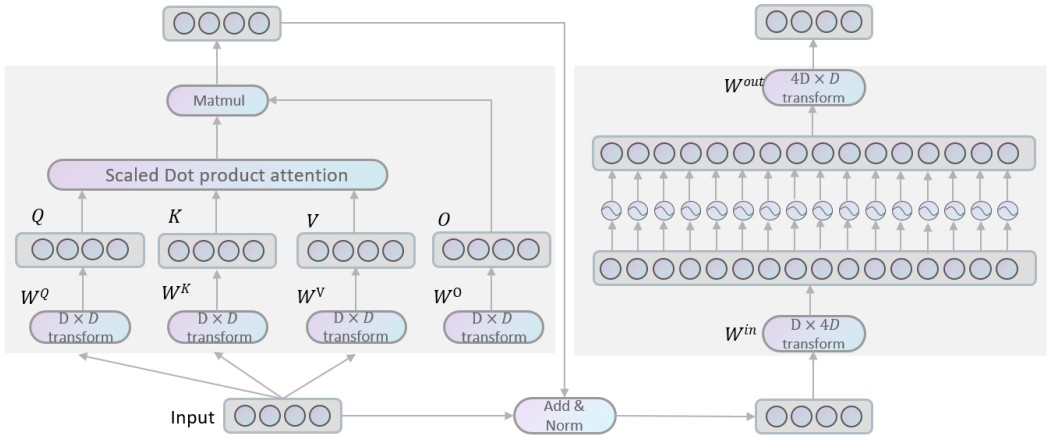

Figure 3: Transformer architecture.

output space $\{Q, K, V\}$ as below [8]:

$$\begin{bmatrix} Q \\ K \\ V \end{bmatrix} = X \times \begin{bmatrix} \boldsymbol{W}^Q \\ \boldsymbol{W}^K \\ \boldsymbol{W}^V \end{bmatrix} \tag{8}$$

The self-attention mechanism (a.k.a Scaled Dot-Product Attention) is calculated as

$$\text{Attention(Q,K,V)} = \text{softmax}\left(\frac{QK}{\sqrt{d_k}}\right)V \tag{9}$$

For a multi-head version of the self-attention mechanism, it linearly projects $Q, K, V$ with $h$ times using individual linear projections to smaller dimensions (e.g. $d_k = \frac{d_{\text{model}}}{h}$), instead of performing a single attention function with $d_{\text{model}}$-dimensional keys, values and queries. Finally, the output of SAN is

$$\begin{aligned} \text{SAN}(X) &= [\text{head}_1; \cdots; \text{head}_h]\boldsymbol{W}^O \\ \text{head}_i &= \text{Attention}(Q_i, K_i, V_i), \end{aligned} \tag{10}$$

where $Q = [Q_1; \cdots Q_h]$, $K = [K_1; \cdots K_h]$, and $V = [V_1; \cdots V_h]$. The individual attention heads are independently calculated, Cordonnier et al. (2021) claims that there is some redundancy in multi-head attention.

Since the output of SAN is a linear transformation (using $\boldsymbol{W}^O$) of $V$, which is a weighted sum of $V$. A stack of many purely SAN layers is not expressive (Dong et al., 2021), since it is equivalent to a single linear transformation. To this end, a feed-forward network with non-linear activation is alternately used with each SAN layer,

$$\text{FFN}(X) = \delta(X\boldsymbol{W}^{\text{in}})\boldsymbol{W}^{\text{out}}. \tag{11}$$

Since some neurons after the activation function (e.g., $\delta$ is ReLU or GELU (Hendrycks & Gimpel, 2016)) become inactivated (zero), $d_{\text{in}}$ is usually bigger than $d_{\text{model}}$ to avoid the low-rank bottleneck, typically, $d_{\text{in}} = 4 \times d_{\text{model}} = d_{\text{out}}$. Other tricks, such as layer normalization, residual connection, dropout, and weight decay are also adopted to relieve the optimization and overfitting problems when it goes deeper.

## A.2 BERT

BERT is a Transformer architecture-based pre-trained language model trained on plain corpora by using a *masked language model* pre-training objective, thanks to the capacity and scalability of the

---

[8]For all linear transformation in this paper, the bias term is in default omitted

Transformer (Devlin et al., 2018). BERT significantly improved the SOTA performance of many downstream tasks, including classification benchmark GLUE (Wang et al., 2018). Note that the parameters of SAN and FAN linearly increase with the number of layers while the embedding layers keeps constant with respect to the layer number.

## B  'MULT-HEAD' FEED FORWARD NEURAL NETWORK

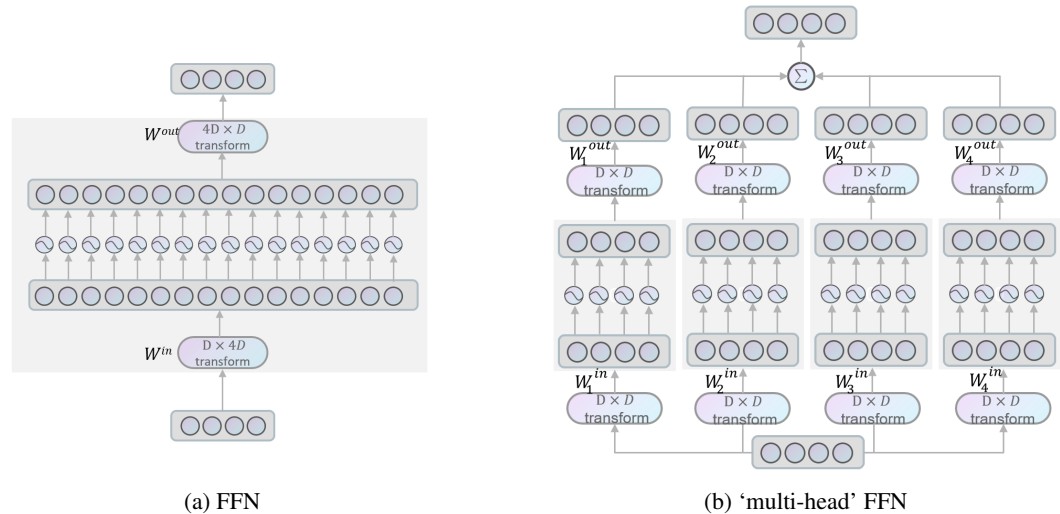

|  (a) FFN  |  (b) 'multi-head' FFN |

Figure 4: Fig. 4a and Fig. 4b are equivalent. Note that FFN could be a sum of 4 independent sub-FFNs; each of them conducts a full-rank $D \times D$ transformation.

In the multi-head attention mechanism, individual attention heads are separately calculated, however the calculation is low-rank due to the redundancy among heads (Cordonnier et al., 2021). In this paper, we argue that such redundancy may also appear in the feed-forward neural networks. In the feed-forward neural networks, element-wise activation functions are usually adopted, e.g., GELU (Hendrycks & Gimpel, 2016) and Relu (Agarap, 2018); that is, each activation can be independently calculated.

By partitioning the original $W^{\text{in}}$ into $H$ column groups and $W^{\text{out}}$ into $H$ row groups, one can revise a feed-forward neural layer (i.e. FFN$(X) = \delta(XW^{\text{in}})W^{\text{out}}$) as a sum of $H$ independent 'thin' sub-FFNs with a dimension of $D_H = 4D/H$ as below:

$$\text{FFN}(X) = \sum_{h=1}^{H} \delta(XW_h^{\text{in}})W_h^{\text{out}} \tag{12}$$

Where $W_h^{\text{in}} \in \mathbb{R}^{D \times D_h}$ and $W_h^{\text{out}} \in \mathbb{R}^{D_h \times D}$, $W^{\text{in}} = [W_1^{\text{in}} \oplus \cdots \oplus W_H^{\text{in}}] \in \mathbb{R}^{D \times 4D}$ and $W^{\text{out}} = [W_1^{\text{out}} \oplus \cdots \oplus W_H^{\text{out}}] \in \mathbb{R}^{4D \times D}$. In this paper, we set $H = 4$, since $W_h^{\text{in}}, W_h^{\text{out}} \in \mathbb{R}^{D \times D}$ and each transformation in sub-FFNs layer are full-rank transformations. See Fig 4 for graphical illustration. One can refer to block partitioned matrix multiplication to understand the equivalence between Fig. 4a and 4b.

## C  CONCATENATION OVER ROWS OR COLUMNS

PCA on columnly-stacked and rowly-stacked matrices are shown in Fig. 5a and Fig. 5b. Observe that there is no too much difference between them.

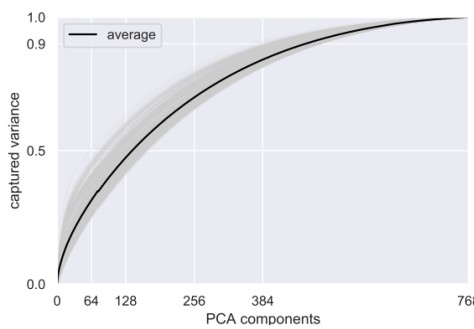 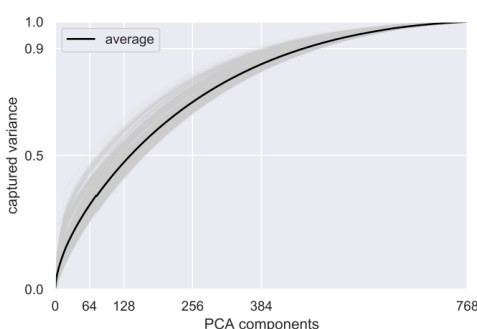

(a) PCA for a pair of matrices along columns    (b) PCA for a pair of matrices along rows

Figure 5: PCA for existing weight block matrices in BERT-base

## D  MATRIX AND TENSOR DECOMPOSITION

Tensor is a generalized 'matrix' with typically a dimension of 3 or more; sometimes, one can also call a matrix as a bi-dimensional 'tensor' and a vector as a one-dimensional 'tensor'. Formally, an $n$-dimensional tensor is an array with $n$ indexes as

$$\mathbf{X} \in \mathbb{R}^{I_1, \cdots, I_n} \tag{13}$$

An entry in $\mathbf{X}$ is accessed by selecting $n$ ordered index $(i_1, \cdots, i_n)$, and $\{0 \le i_k < I_n, i_k \in \mathbb{N}\}$. In this section, we mainly discuss order-3 tensors for simplicity while higher-order tensors also hold.

Typical tensor decomposition includes Canonical Polyadic (CP) decomposition (Carroll & Chang, 1970; Harshman, 1972) and Tucker decomposition (Tucker, 1963). CP decomposition approximates a high-dimension tensor with as a sum of many rank-one tensors. In the case of three-dimensional tensor decomposition,

$$\hat{\mathbf{X}} = \sum_{r=1}^{R} \boldsymbol{A}_r \otimes \boldsymbol{B}_r \otimes \boldsymbol{C}_r \tag{14}$$

Where $\boldsymbol{A}_r \in \mathbb{R}^{I_1}$, $\boldsymbol{B}_r \in \mathbb{R}^{I_2}$, $\boldsymbol{C}_r \in \mathbb{R}^{I_3}$ and $\boldsymbol{A} \in \mathbb{R}^{R \times I_1}$, $\boldsymbol{B} \in \mathbb{R}^{R \times I_2}$ and $\boldsymbol{C} \in \mathbb{R}^{R \times I_3}$. $\otimes$ is the tensor product [9].

Tucker decomposition decomposes a tensor into a set of factor matrices and one small low-rank core tensor,

$$\hat{\mathbf{X}} = \mathbf{G} \times_1 \boldsymbol{A} \times_2 \boldsymbol{B} \times_3 \boldsymbol{C} \tag{15}$$

where $\boldsymbol{A} \in \mathbb{R}^{R_1 \times I_1}, \boldsymbol{B} \in \mathbb{R}^{R_2 \times I_2}, \boldsymbol{C} \in \mathbb{R}^{R_3 \times I_3}$ and $\mathbf{G} \in \mathbb{R}^{R_1 \times R_2 \times R_3}$. $\times_1$, $\times_2$, and $\times_3$ are mode-$k$ products [10]. $\{R_1, R_2, R_3\}$ is sometimes called *Tucker ranks*. An entry with index $(i_1, i_2, i_3)$ is calculated as

$$\hat{\mathbf{X}}_{i_1, i_2, i_3} = \sum_{a=1}^{R_1} \sum_{b=1}^{R_2} \sum_{c=1}^{R_3} \mathbf{G}_{a,b,c} \boldsymbol{A}_{a,i_1} \boldsymbol{B}_{b,i_2} \boldsymbol{C}_{c,i_3} \tag{16}$$

---

[9] Here we give an example to calculate the tensor product of three 2-dimensional vectors, $\boldsymbol{x}, \boldsymbol{y}, \boldsymbol{z} \in \mathbb{R}^2$, resulting in a tensor of $\mathbb{R}^{2 \times 2 \times 2}$. R is the *CP rank* shared in all modes.

$$\mathbf{x} \otimes \mathbf{y} \otimes \mathbf{z} = \begin{bmatrix} x_1 \\ x_2 \end{bmatrix} \otimes \begin{bmatrix} y_1 \\ y_2 \end{bmatrix} \otimes \begin{bmatrix} z_1 \\ z_2 \end{bmatrix} = \begin{bmatrix} x_1 \\ x_2 \end{bmatrix} \otimes \begin{bmatrix} y_1 z_1 & y_1 z_2 \\ y_2 z_1 & y_2 z_2 \end{bmatrix} = \begin{bmatrix} \begin{bmatrix} x_1 y_1 z_1 \\ x_1 y_1 z_2 \\ x_2 y_1 z_1 \\ x_2 y_1 z_2 \end{bmatrix} \begin{bmatrix} x_1 y_2 z_1 \\ x_1 y_2 z_2 \\ x_2 y_2 z_1 \\ x_2 y_2 z_2 \end{bmatrix} \end{bmatrix}$$

[10] Given a tensor $\mathbf{G} \in \mathbb{R}^{R_1 \times \mathbb{R}_2 \cdots R_k \cdots \mathbb{R}_n}$ and a matrix $\boldsymbol{M} \in \mathbb{R}^{R_k \times r}$, a mode-$k$ between $\mathbf{G}$ and $M$ results in $\mathbf{G} \times_m \boldsymbol{M} \in \mathbb{R}^{R_1 \times R_2 \cdots R_{k-1} \times r \times R_{k+1}, \cdots, R_n}$.

Tucker decomposition will degrade to CP decomposition,where the core tensor **G** is constrained to be super-diagonal [11] and $R = R_1 = R_2 = R_3$.

A $k$-slice in the first mode, i.e., a matrix with a size of $I_2 \times I_3$, would be

$$\hat{\mathbf{X}}_k = \sum_{i_1=1}^{R_1} \sum_{i_1=1}^{R_2} \sum_{i_3=1}^{R_3} \mathbf{G}_{i_1,i_2,i_3} \boldsymbol{A}_{i_1,k} (\boldsymbol{B}_{i_2} \otimes \boldsymbol{C}_{i_3}), \quad \hat{\mathbf{X}}_k \in \mathbb{R}^{I_2 \times I3} \tag{17}$$

All slices in a specific mode share factor matrices (i.e., $\boldsymbol{B}, \boldsymbol{C}$) in other modes.Therefore, there exists inter-correlation between these slices. These shared factor matrices not only make the compression ratio of *tensor decomposition* being much bigger than multiple independent *matrix decomposition* (Noach & Goldberg, 2020), but also can be beneficial especially when there is redundancy among these parameter matrices and factor matrices can be utilized common features.

# E    DIFFERENCE WITH EXISTING WORK

## E.1    DIFFERENCE WITH TENSOR DECOMPOSITION USED FOR COMPRESSING CNNS, RNNS AND EMBEDDINGS

Compressing pre-training models is a new scenario. We believe that exploring tensor decomposition in pre-trained language models is non-trivial. In the pre-trained language models, we could test tensor decomposition in very deep and wide networks like GPT 3 (96 layers and a hidden dimension of 12288), while this work is the first step.

Existing work Ma et al. (2019); Liu et al. (2021); Gao et al. (2020); Khrulkov et al. (2019); Hrinchuk et al. (2020); Panahi et al. (2019) which use tensor network for compressing neural networks do not have the potential for acceleration. The bottleneck in speed limits the application of tensor decomposition in real-world applications: compressing models but consuming longer inference time seems be useful in very rare scenarios. We argue that it is nontrivial to compress neural networks using tensor decomposition with acceleration effects, as this work did.

Work mechanisms for compression are totally different, previous works compress each weight matrix ($\boldsymbol{W}^Q, \boldsymbol{W}^K, \boldsymbol{W}^V, \boldsymbol{W}^O, \boldsymbol{W}^{in}, \boldsymbol{W}^{out}$ in each layer) individually using matrix/tensor decomposition. They are making use of local redundancy inside each matrix. While in big models (PLMs), we believe that making use of cross-matrix/cross-layer redundancy is also, sometimes more, beneficial. We believe that using tensor decomposition for cross-matrix/cross-layer redundancy is a significant difference.

## E.2    DIFFERENCE WITH CORDONNIER ET AL. (2021)

(Cordonnier et al., 2021) is quite impressive and inspiring. It does inspire this paper, however, we want to highlight the difference with (Cordonnier et al., 2021).

**Motivation**    The motivation is different, (Cordonnier et al., 2021) found the redundancy in different heads. We make use of redundancy of both intra-matrix redundancy and inter-matrix redundancy (including cross-layer redundancy).

**Method**    The architecture in (Cordonnier et al., 2021) is slightly different from Transformer (or BERT) since it redesigns the self-attention network with collaborated attention. While our architecture is nearly the same as the raw model, we simulate each matrix multiplication of BERT with a product of many smaller matrices.

**Goal**    Our work aims to extremely compress neural networks, which cannot be achieved by (Cordonnier et al., 2021). (Cordonnier et al., 2021) is to explore the possibility to share key/query projection in SANs. Note that SAN only has 1/3 parameters in Transformer, this proportion even becomes smaller when considering the embedding layer. By compressing only SANs with 1/3 parameters, its overall compression ratio is limited.

---

[11]Namely, $\mathbf{G}_{i_1,i_2,i_3}$ equals 1 if $i_1 = i_2 = i_3$, and 0 otherwise

**Potential for efficiency** (Cordonnier et al., 2021) is faster than the raw model only if $D_k$ is small. The smallest setting with $D_k = 64$ has **20%** FLOPs reduction. This shows its potential for acceleration is limited. A similar setting with $D_k = 64$, we have nearly **80%** reduction in terms of FLOPs.

**Potential for compression ratios** The smallest model for BERT-base (110M) in (Cordonnier et al., 2021) has **96.6M** when $D_k = 128$; its best compression ratio is 1.14. While in our model, the smallest model has a much bigger compression ratio, while being faster and performing better.

**Potential for effectiveness** The model in Cordonnier et al. (2021) (**96.6M** parameters) achieves **93.5%** (77.6/83.0) performance with BERT base, while our smallest model with **25M** parameters (plus parameters in the embedding layer) achieves **97.7%** performance (80.8/82.7) of BERT base. A slight difference is that we test our model on the test dataset through GLUE online benchmark while Cordonnier et al. (2021) test their model on offline dev dataset through average results for three runs, so we use the relative performance w.r.t. the original BERT-base.

## F    MORE DISCUSSIONS TO COMPRESS WEIGHTS MATRICES IN SANs AND FFNs TOGETHER

### F.1    SHARING LEADS TO A BIGGER MATRIX BANK

If we separately compress SANs and FFNs, we would have two matrix banks: one for SANs and one FFNs: $\boldsymbol{P}^{FFN}\mathbf{C}^{FFN}$ and $\boldsymbol{P}^{SAN}\mathbf{C}^{SAN}$. Each weight matrix in FFN(or SAN) is specific to a matrix bank $\mathbf{C}^{FFN}$ for FFN (or $\mathbf{C}^{SAN}$ for SAN) and its weight vector over the bank $\boldsymbol{P}_i^{FFN}$ (or $\boldsymbol{P}_i^{SAN}$). Note that the two matrix banks have the most parameters since it is three-order tensor while others $(\boldsymbol{U}, \boldsymbol{V}, \boldsymbol{P})$ are matrices.

Note that matrices in two matrix banks have the same shape, one could merge (share) the two matrix banks (a $m$-size $d \times d$ matrix bank and a $n$-size $m$-size $d \times d$ matrix bank) to get a single bigger $((m + n)$-size) matrix bank , this could boost the expressive power for both FFNs and SANs due to the bigger matrix bank (denoted as $[\mathbf{C}^{FFN}; \mathbf{C}^{SAN}]$).

The unmerged one is a special case of the merged one. Let us define a new $\boldsymbol{P}'$, each element in which is defined as below:

$$\boldsymbol{P}_i' = \begin{cases} \left[\boldsymbol{P}_i; [\overbrace{0, 0, \cdots 0}^{n}]\right] \text{ for SANs} \\[2em] \left[[\overbrace{0, 0, \cdots 0}^{m}]; \boldsymbol{P}_i\right] \text{ for FFNs} \end{cases}$$

$\boldsymbol{P}_i'$ is the new weight vector over the shared $(m + n)$-size matrix bank.

Without losing any generality, we could relieve the zero-constrains in $\boldsymbol{P}'$ to get a general form that each element in $\boldsymbol{P}'$ is not forced to be zero. This could make FFNs and SANs share more matrix bases and therefore get better capacity.

The benefit could be empirically evidenced in Table 7: solely compressing SANs (without compressing FFNs) underperforms both compressing FFNs and SANs; although the former has much more parameters. Since in the shared setting, the shared matrix bank could compensate the counterpart of SANs. Another naïve motivation is to design a unified protocol for both SANs and FFNs.

### F.2    DISCUSSIONS ON THE SCALE DIFFERENCE BETWEEN WEIGHTS MATRICES IN SANs AND FFNs

We want to clarify that we do not intend to approximate the raw weights (and their norm/scale). Previously, we tried to add a regularizer in the training loss function to force the reconstructed weights using decomposition to be as close as the raw weights.

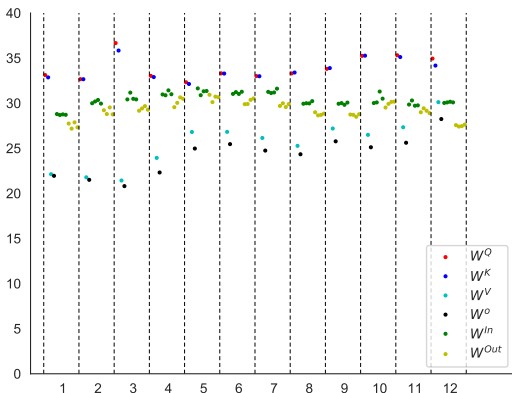

Figure 6: L2 norms of raw weights in BERT

$$\mathcal{L} = \mathcal{L}_{\text{training}} + \lambda |\mathbf{W}^{IV} - \mathbf{W}^{I}|_2$$

This does not improve the performance, but worsen the performance. The reason may be as below. Since we cannot perfectly approximate the raw weights with a decent compression ratio, there always exists some difference between the raw weights and the reconstructed weights using decomposition. However, even a slight difference might lead to a big difference in output. Plus, we also observe that the finally learned $\mathbf{W}^{IV}$ has a big difference with $\mathbf{W}^{I}$. So we give up our efforts to approximate the raw weights.

For example, we found that even randomly initializing the compressed BERT nearly achieves identical performance, compared to decomposition from a given model. This also shows that approximating raw weights is not beneficial. Instead, we use knowledge distillation to simulate the input-output mapping from the original model and tolerate the difference between raw weights and reconstructed weights. We believe that the former makes more sense than the latter. Empirical results show the former performs well.

**Norm difference in decomposition** Decomposition is to approximate a three-order tensor with three factor tensors and a core tensor.

$$\mathbf{W}_i = (\mathbf{C}P_i) \times_2 U \times_3 V$$

Each matrix has its specific factor vector $P_i$, which does not have any constraints. The flexibility of norms in $P_i$ could compensate the norm difference for the original matrices.

In case the readers may be curious, we show the norms in Figure 6. The norms in different matrices do have some differences. We also compare the norms between the raw weight matrices and the reconstructed weights matrices in Figure 7. It shows that the norms of reconstructed weights also have a big difference compared to that of corresponding original weight matrices. We argue that the knowledge distillation does not aim to approximate the raw weight matrices but only approximate its input-output mapping functions.

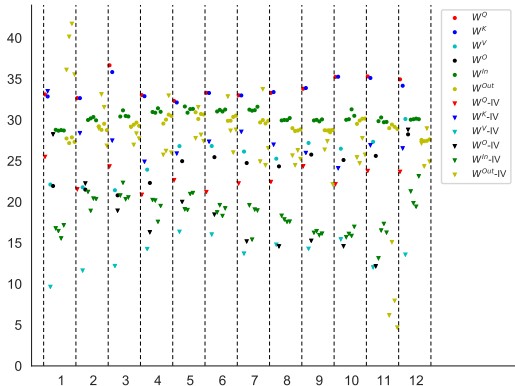

Figure 7: L2 norms of raw weights in BERT and reconstructed BERT-IV.

## G  LINKING IV TO ALBERT

For IV, we have $\mathbf{W}^{\text{IV}} = U(PC)V$. Considering a specific case when $U = V = I_{(D)}$, $l = 12$ (i.e.,

$C \in \mathbb{R}^{12 \times D^2}$) and $P = \begin{bmatrix} I_{(12)} \\ I_{(12)} \\ \cdots \\ I_{(12)} \end{bmatrix} \in \mathbb{R}^{12L \times 12}$. Where $I_{(i)}$ is an $i \times i$ identity matrix. Namely

$$I_{(i)} = \begin{bmatrix} 1 & 0 & \cdots & 0 \\ 0 & 1 & \cdots & 0 \\ \cdots & & & \\ 0 & 0 & \cdots & 1 \end{bmatrix} \in \mathbb{R}^{i \times i}. \tag{18}$$

Then

$$\begin{aligned} W^{\text{IV}} &= U(P_i C)V \\ &= PC \\ &= \begin{bmatrix} C \\ C \\ \cdots \\ C \end{bmatrix} \in \mathbb{R}^{12LD^2} \end{aligned} \tag{19}$$

Eq. 19 results in a layer-shared BERT like ALBERT (Lan et al., 2019). (Lan et al., 2019) additionally compresses the embedding layer using matrix factorization.

## H  HOW IV GENERALIZES III

*III (tensor train decomposition) is a special case of IV (tucker decomposition) in this paper.* Note that, we are not stating the correspondence between tensor train decomposition and tucker decomposition in general case, but a three-order tensor train decomposition is a special case of tucker decomposition.

Let us recall their definition:

$$\begin{cases} \mathbf{W_i}^{III} = U\Sigma_{\mathbf{i}}V \\ \mathbf{W_i}^{IV} = U(P_i C)V \end{cases}$$

Or in another format:

$$\begin{cases} \mathbf{W}^{III} = \Sigma \times_2 U \times_3 V \\ \mathbf{W}^{IV} = (PC) \times_2 U \times_3 V \end{cases}$$

$\times_2$ and $\times_3$ are the mode-2 and mode-3 multiplication. The only difference $\Sigma \in \mathbb{R}^{12L \times d \times d}$ vs. $(PC) \in \mathbb{R}^{12L \times d \times d}$. In the latter, $P \in \mathbb{R}^{12L \times l}$ and $C \in \mathbb{R}^{l \times d \times d}$. In a special case of tucker

Table 8: Task descriptions and statistics in GLUE (Wang et al., 2018). NLI is for 'Natural Language Inference' and QA is for 'Question Answering'. SST-2, MNLI, QNLI, QQP are considered as relatively-big dataset according to the scale of their train set.

|  | CoLA | SST-2 | MNLI | MRPC | QNLI | QQP | RTE | STS-B |
|---|---|---|---|---|---|---|---|---|
| task | classification | classification | NLI | paraphrase | QA/NLI | paraphrase | NLI | similarity |
| train | 8.5 k | 67k | 393k | 3.7k | 105k | 364k | 2.5k | 7k |
| test | 1k | 1.8k | 20k | 1.7k | 5.4k | 391k | 3k | 1.4k |

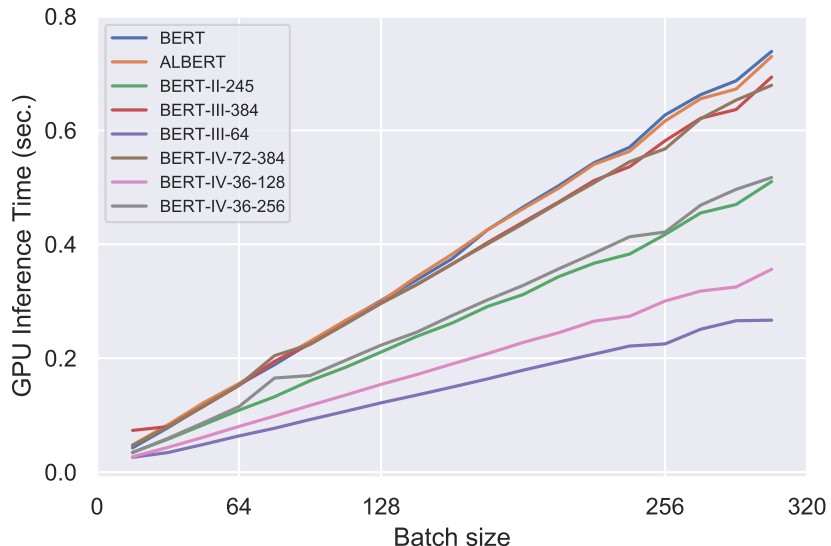

Figure 8: Inference time. Inference time in Y-axis (in a Nvidia V100 16G GPU) increases when batchsizes (in X-axis) become bigger. BERT-III/BERT-IV with $d < 384$ are faster than raw BERT.

decomposition when 1) $12L = l$ and 2) $P = I_{12L}$, it is equivalent to the 3-order tensor train decomposition (III). $I_{12L}$ is a (square) identity matrix with a size of $12L \times 12L$, in which the diagonal element equals one and 0 elsewhere.

In our experiments, there existed two settings which set $12L = l = 144$ that satisfied 1). In such settings, the rigorous difference with BERT-III and BERT-IV is that we do not force $P$ to be equal to an identity matrix, instead, $P$ is flexible to be trained. We argue that adding such a flexible $P$ could enable crossing layer sharing and enhance the capacity of III. Interestingly, once we finish training BERT-IV-144-384 and BERT-IV-144-64, before starting inference, one could merge $P$ and and **C** as a single matrix: $\mathbb{R}^{12L \times 12L} \times \mathbb{R}^{12L \times d^2} \to \mathbb{R}^{12L \times d^2}$, which does not affect the model but slightly improve inference speed. By doing so, it is rigorously equivalent to III. Thus, we rename BERT-IV-144-384/BERT-IV-144-64 as BERT-III-384 and BERT-III-384.

# I  GLUE BENCHMARK

The data statistics in GLUE is shown in Tab. 8.

# J  INFERENCE TIME

As shown in Fig. 8, the raw BERT base is the slowest one. BERT-III/BERT-IV with dimension rank $d = 384$ is slight faster than raw BERT. The inference time consistently decreases when $d$ becomes smaller. Especially, BERT-III-64 is 2.6 × faster than BERT, which is consistent to RPS shown in Tab. 4.

Note that BERT-III -384 and BERT-IV -72-384 are faster than BERT in terms of inference time and RPS (see RPS in Tab. 4), although they have the identical FLOPS. A similar observation could be found in ALBERT (Lan et al., 2019) – layer-shared BERT has the same FLOPS with the raw BERT while the former is faster than the latter. The better inference time and RPS with the same FLOPS may be due to that the relatively smaller model consumes less memory and bandwidth inside GPU, see `https://openreview.net/forum?id=H1eA7AEtvS`. This speedup effect in this matter depends on specific platforms and hardware.

BERT-II implemented by Noach & Goldberg (2020) has a rank for hidden dimension of 245, there we called 'BERT-II-245'. It inference time is close to our BERT-IV-36-256, this is because the inference time is mainly related to the dimension rank $d$.

## K   ON THE COMPRESSION OF THE EMBEDDING LAYER

In this work, we did not compress the embedding layer. The reasons to not compress the embedding layer are manyfold. First, compressing the embedding layer will definitely increase training/inference time, since a single embedding lookup is fast enough.

Secondly, the embedding layer does not increase the total parameters when BERT has more transformer layers in the encoder. Since the parameters in the embedding layer are constant with respect to network depth. Table 9 shows the parameter numbers when compressing main weight matrices with a half dimension rank (when the hidden dimension is 768, its half is 384 as the rank d, meanwhile keeping layer rank unchanged). Here, we also consider embedding layer for a fair comparison as suggested.

| models | # Paras. | # layers (L) | D | V | BERT-IV-$\frac{12L}{2} - \frac{D}{2}$ | compression ratios |
|---|---|---|---|---|---|---|
| BERT-base-uncased | 110M | 12 | 768 | 30522 | 35.7M | 3.1 |
| BERT-large-uncased | 340M | 24 | 1024 | 30522 | 75.8M | 4.5 |
| GPT-3 Small | 125M | 12 | 768 | 50257 | 50.7M | 2.5 |
| GPT-3 Medium | 350M | 24 | 1024 | 50257 | 85.8M | 4.1 |
| GPT-3 Large | 760M | 24 | 1536 | 50257 | 165.5M | 4.6 |
| GPT-3 XL | 1.3B | 24 | 2048 | 50257 | 243.0M | 5.3 |
| GPT-3 2.7B | 2.7B | 32 | 2560 | 50257 | 498.0M | 5.4 |
| GPT-3 6.7B | 6.7B | 32 | 4096 | 50257 | 1.1B | 6.3 |
| GPT-3 13B | 13.0B | 40 | 5140 | 50257 | 1.9B | 6.8 |
| GPT-3 175B | 175.0B | 96 | 12288 | 50257 | 22.8B | 7.7 |

Table 9: Compression ratios with increasing models when consider parameters in the embedding layer. The bigger the model, the closer it is to the theoretical upper bound of compression ratios (i.e., 8).

Note that the parameters of embedding becomes more negligible when PLMs have more layers and bigger hidden dimension, in which case the compression ratio will approximate an ideal upper bound ($8 = 2 \times 2^2$, which is linear to the deduction times of *layer rank* and quadratic to the deduction times of *dimension rank*; in practice, we could use bigger deduction in both ranks, as we did in the experiments).

Plus, the shape of an embedding layer ($VD$) is related to the size of vocabulary, which is heterogeneous to other weight matrices in Self-attention network and Feed-forward network ($D^2$ or $4D^2$). Therefore it is incompatible with the current compression protocol. To additionally compress the embedding layer, we might have to design a totally different compression protocol for the embedding layer, which makes this work more complicated.

Finally, the embedding layer is relatively easy to compress, see tensorized embedding (Hrinchuk et al., 2020) and word2ket (Panahi et al., 2019) which compress embedding with maximum 307 and 93,675 times respectively with a slight performance drop. We believe that it is trivial to additionally compress the embedding layer. Thus, we leave compressing the embedding layer as future work.

Table 10: Experimental comparisons between the proposed work and (Noach & Goldberg, 2020). & indicates the reported number is an average between these metrics. Metrics are Accuracy (MNLI (average of MNLI match and MNLI mis-match), QNLI, RTE, SST-2), Avg of Accuracy and F1 (MRPC, QQP), Matthew's correlation (CoLA), Avg of Pearson and Spearman correlations (STS-B).

| Model | Para. | FLOPS | RPS | SST-2 acc | MNLI acc (m&mm) | MRPC acc&F1 | QNLI acc | QQP acc (m&mm) | RTE acc | STS-B spear.&pears. | Avg |
|---|---|---|---|---|---|---|---|---|---|---|---|
| BERT-base | 86.0M | 22.5B | 420.1 | **93.4** | 83.1 | 85.6 | **90.9** | 80.2 | 66.4 | **86.5** | 83.9 |
| BERT-III -384 | 23.0M | 22.5B | 452.2 | **93.4** | **84.2** | **86.2** | 90.5 | 80.5 | 68.1 | 86.1 | **84.0** |
| BERT-III-64 | 1.8M | 4.3B | 1088.6 | 91.9 | 79.9 | 85.5 | 82.3 | 79.8 | 63.3 | 81.3 | 80.6 |
| BERT-IV-72-384 | 12.3M | 22.5B | 452.3 | 93.1 | 83.6 | 86.0 | 90.2 | 80.4 | 67.3 | 85.6 | 83.7 |
| BERT-IV-36-256 | 3.9M | 15.2B | 596.9 | 92.7 | 82.2 | 84.4 | 88.9 | 80.2 | 65.2 | 82.4 | 82.3 |
| BERT-IV-36-128 | 1.9M | 8.0B | 863.0 | 92.4 | 80.7 | 86.5 | 83.2 | **80.6** | 64.4 | 82.0 | 81.4 |
| matrix decomposition (Noach & Goldberg, 2020) ( BERT-II-245) | 41.8M | 14.6B | 656.8 | 91.9 | 79.9 | 85.5 | 82.3 | 79.8 | 63.3 | 81.3 | 80.6 |

Table 11: Experimental comparisons between the proposed work and (Lagunas et al., 2021).

| Model | Para. | FLOPS | RPS | SST-2 acc | MNLI acc (m&mm) | QQP f1 | Avg |
|---|---|---|---|---|---|---|---|
| BERT-base | 86.0M | 22.5B | 420.1 | 92.5 | 84.5/84.9 | 87.7 | 87.4 |
| BERT-III -384 | 23.0M | 22.5B | 452.2 | **92.6** | **86.7/86.7** | **88.0** | **88.5** |
| BERT-III-64 | 1.8M | 4.3B | 1088.6 | 91.4 | 80.7/80.8 | 87.2 | 85.0 |
| BERT-IV-72-384 | 12.3M | 22.5B | 452.3 | 92.5 | 85.6/85.6 | 87.7 | 87.9 |
| BERT-IV-36-256 | 3.9M | 15.2B | 596.9 | 92.2 | 83.4/83.8 | 88.0 | 86.9 |
| BERT-IV-36-128 | 1.9M | 8.0B | 863.0 | 91.6 | 82.7/82.6 | 87.5 | 86.1 |
| Sparsified BERT (Lagunas et al., 2021) | 20M | - | - | 91.2 | 83.2/83.6 | 87.9 | 86.5 |

## L    COMPARISON TO METHODS THAT USES NON-STANDARD METRICS IN GLUE

(Noach & Goldberg, 2020) uses non-standard setting for GLUE tasks. Metrics in (Noach & Goldberg, 2020) are Accuracy (MNLI (average of MNLI match and MNLI mis-match), QNLI, RTE, SST-2), Avg of Accuracy and F1 (MRPC, QQP), Matthew's correlation (CoLA), Avg of Pearson and Spearman correlations (STS-B). Tab. 10 tailored our result in their evaluation protocol. In this work, we do not include CoLA dataset since training samples in CoLA are too few to have stable evaluations.

## M    ANALYSIS OF THE LEARNED MODEL

Observing that in IV, each matrix is parameterized as $\mathbf{W}_i^{\mathrm{IV}} = U(P_i\mathbf{C})V$. With shared $U$ and $V$, each matrix has a specific parameter $P_i$ (called 'factor vectors' later), we analyze $\{P_i\}$ to shed some light on understanding what BERT-IV learns.

In BERT-IV-72-384, we calculate the cosine distances between any two factor vectors. $d(P_i, P_j) = 1 - \cos(P_i, P_j)$, lead to a $144 \times 144$ matrix. We could observe that there is a cross-layer pattern that $\{W^{In}\}$ (the weight matrices in the layer of FFNs) among layers are relatively close. This pattern becomes weaker in deep (top) layers, this might suggest that top layers may learn more diverse feature projections. The reason for this pattern needs further investigation.

We also visualize the in-matrix pattern in each layer (e.g. in the 1st, 4-th, 8-th, and 12-th layer) in Figure 10. It shows that a pair between $W_h^{In}$ and $W_h^{Out}$ is relatively close, e.g., $W_1^{In}$ and $W_1^{Out}$, $W_2^{In}$ and $W_2^{Out}$, etc.

## N    POTENTIAL OF COMPRESSION IN LAGER MODELS

In this section, we compare various compression ratios of II, III, and IV, see the parameters and their compression ratios (this additionally considers parameters in the embedding layer) in Table 12. It shows that III and IV could achieve much bigger compression ratios than II when using the same rank for hidden dimension (i.e., 2048). We claim that the proposed model has better potential to compress bigger models.

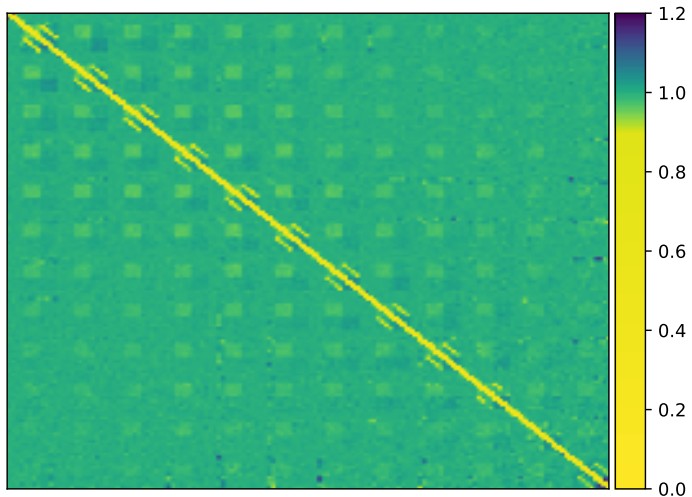

Figure 9: Distances between the factor vectors among 144 matrices in a trained BERT-IV-72-384. The order is listed as $\left[ \boldsymbol{W}^Q, \boldsymbol{W}^K, \boldsymbol{W}^V, \boldsymbol{W}^O, \boldsymbol{W}_1^{In}, \boldsymbol{W}_2^{In}, \boldsymbol{W}_3^{In}, \boldsymbol{W}_4^{In}, \boldsymbol{W}_1^{Out}, \boldsymbol{W}_2^{Out}, \boldsymbol{W}_3^{Out}, \boldsymbol{W}_4^{Out} \right]$ from the first layer to last layer.

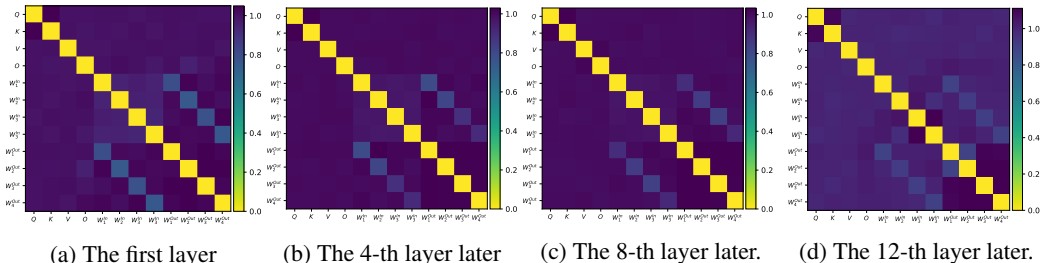

| (a) The first layer | (b) The 4-th layer later | (c) The 8-th layer later. | (d) The 12-th layer later. |

Figure 10: Distances between the factor vectors among 12 matrices in each layer of a trained BERTIV-72-384. The order is listed as $\left[ \boldsymbol{W}^Q, \boldsymbol{W}^K, \boldsymbol{W}^V, \boldsymbol{W}^O, \boldsymbol{W}_1^{In}, \boldsymbol{W}_2^{In}, \boldsymbol{W}_3^{In}, \boldsymbol{W}_4^{In}, \boldsymbol{W}_1^{Out}, \boldsymbol{W}_2^{Out}, \boldsymbol{W}_3^{Out}, \boldsymbol{W}_4^{Out} \right]$.

| model | Paras | $L$ | $D$ | GPT-II-2048 | GPT-III-2048 | GPT-IV-12-768 | GPT-IV-12-2048 | GPT-IV-144-2048 |
|---|---|---|---|---|---|---|---|---|
| GPT-3 Small | 125M | 12 | 768 | 493.1M (0.3× ↓) | 647.2M (0.2× ↓) | 48.3M (2.6× ↓) | 93.5M (1.3× ↓) | 647.2M (0.2× ↓) |
| GPT-3 Medium | 350M | 24 | 1024 | 1.3B (0.3× ↓) | 1.3B (0.3× ↓) | 56.7M (6.2× ↓) | 102.5M (3.4× ↓) | 656.2M (0.5× ↓) |
| GPT-3 Large | 760M | 24 | 1536 | 1.9B (0.4× ↓) | 1.3B (0.6× ↓) | 90.0M (8.4× ↓) | 137.1M (5.5× ↓) | 690.8M (1.1× ↓) |
| GPT-3 XL | 1.3B | 24 | 2048 | 2.5B (0.5× ↓) | 1.3B (1.0× ↓) | 102.3M (12.7× ↓) | 150.8M (8.6× ↓) | 704.5M (1.8× ↓) |
| GPT-3 2.7B | 2.7B | 32 | 2560 | 4.2B (0.6× ↓) | 1.8B (1.5× ↓) | 194.4M (13.9× ↓) | 244.2M (11.1× ↓) | 797.9M (3.4× ↓) |
| GPT-3 6.7B | 6.7B | 32 | 4096 | 6.7B (1.0× ↓) | 1.9B (3.6× ↓) | 270.9M (24.7× ↓) | 324.7M (20.6× ↓) | 878.4M (7.6× ↓) |
| GPT-3 13B | 13.0B | 40 | 5140 | 10.4B (1.2× ↓) | 2.4B (5.5× ↓) | 333.6M (39.0× ↓) | 390.0M (33.3× ↓) | 943.7M (13.8× ↓) |
| GPT-3 175B | 175.0B | 96 | 12288 | 59.0B (3.0× ↓) | 5.9B (29.5× ↓) | 1.1B (**162.1×** ↓) | 1.2B (151.6× ↓) | 1.7B (102.4× ↓) |

Table 12: Parameter compression ratios in various models