# OpenReview forum: "Exploring extreme parameter compression for pre-trained language models"
_ICLR.cc/2022/Conference — ICLR 2022 Poster_

### Official Review · Reviewer_7TPf · 2021-10-23

**Correctness:** 2
**Technical Novelty And Significance:** 2
**Empirical Novelty And Significance:** 3
**Recommendation:** 6
**Confidence:** 4

**Main Review:**

Strengths:
- The paper is well written and the motivating analyses (e.g. Fig 1) are interesting. I also appreciated the thorough appendix.
- The main technical contributions of the paper (i.e. using all the matrices to perform matrix decomposition, using a bank of matrices) is novel to my knowledge. However, I have some reservations about whether this is sound (see weaknesses).

Weaknesses:
- The paper introduces many variants of decomposing the matrices (Table 2). However the results only seem to be based on one of the methods (i.e. IV). I realize that some of variants have already been studied in the literature (e.g. Mao et al., Noach and Goldberg), but since the setup is not identical, it is crucial that the proposed approach is compared against both II and III. Therefore, it is not clear whether the improvement in performance is coming from the actual proposed method, or something else.
- Since the matrices across layers often have different scaling, and since tensor decomposition is approximating some reconstruction error (i.e. L2 in the case of SVD), it's not clear that performing decomposition with all the matrices makes sense.
- The method requires a pipelined approach where one must first perform generalized distillation against the BERT model before doing task-specific distillation.

Questions:
- In GD, did you try distilling also the masked LM logits, in addition to (or instead of) the last layer hidden states and attention maps?
- Have you checked the norms of the matrices? Are they of similar scales? If not, do you obtain better performance by normalizing the matrices such that they are in the same scale?


**Summary Of The Paper:**

After author rebuttal: thank you very much for the detailed response to both my and the other reviewers' comments. I have updated my score.
------------
This paper proposes an approach to compressing the Transformer family of pretrained language models via tensor decomposition. Compared to existing work, the main differences are: (1) performing global tensor decomposition which takes into account matrices across different layers, and (2) using a matrix bank to enable even greater model compression. The approach, when combined with distillation, outperforms existing methods for compressing BERT models on the popular GLUE benchmark.


**Summary Of The Review:**

A new approach to model compression with strong-ish empirical results. Some reservations about the soundness of the method. And it is furthermore not clear that the improvements are coming from the proposed approach as opposed to something else, since an ablation study across the different decomposition methods is missing.

---

> ### Author Response · Authors · 2021-11-15
> **Replies to Reviewer 7TPf**
>
> **R 4.1 Test TT and MF**
>
> Sorry for crossing positing with R 2.1
>
> Thanks for your interest in tensor train decomposition, denoted as III. We understand the concern from the reviewer about tensor train decomposition as experimental comparisons.
>
>
> Actually, **III (tensor train decomposition) is a special case of IV (tucker decomposition) in this paper**. More interestingly, we have implicitly implemented  III in the experiments; this was even not realized by ourselves  when submitting this paper. In the revised paper, we will not highlight only  Tucker decomposition; instead, we exhaustively implement  and investigate III  and IV.
>
> *why III (tensor train decomposition) is a special case of IV (tucker decomposition)* Note that we are not stating the correspondence between tensor train decomposition and tucker decomposition in the general case, but a three-order tensor train decomposition is a special case of tucker decomposition.
>
> $
> \begin{cases}
> \mathbf{W_i}^{III} = U \mathbf{\Sigma_i} V \\
> \mathbf{W_i}^{IV} = U (P _i \mathbf{C})  V  \\
> \end{cases}
> $
>
> Or in the other format:
> $
> \begin{cases}
> \mathbf{W}^{III} =  \mathbf{\Sigma}  \times _2 U \times _3 V  \\
>  \mathbf{W}^{IV} = (P  \mathbf{C}) \times _2 U \times _3 V  \\
>  \end{cases}
> $
>
> $\times _1$ and $\times _3$ are the mode-$1$ and mode-$3$ multiplication.
>
> The only difference between   $\Sigma \in  \mathbb{R}^ { 12L \times  d\times d}$ and $(P  \mathbf{C}) \in  \mathbb{R}^{ 12L \times  d\times d}$. In the latter,  $P  \in  \mathbb{R}^{ 12L \times l}$ and $\mathbf{C} \in \mathbb{R}^{ l \times  d\times d} $. In a special case of tucker decomposition when (1) $12L= l$ and (2) $P = I_{12L}$, it is equivalent to the 3-order tensor train decomposition (III). $ I _{12L}$ is an (square) identity  matrix with a size of $12L \times 12L$, in which the diagonal element equals to one and 0 elsewhere.
>
> In our experiments, there existed two settings previously called TuckerBERT-144-384 and TuckerBERT-144-64, which set  $12L= l = 144$ that satisfied (1). The rigorous difference with   TuckerBERT-144-384/TuckerBERT-144-64 and III is that we do not force $P$ to be equal to a identity matrix, instead $P$ is flexible to be trained. We argue that adding such a  flexible $P$ could enable crossing layer sharing and enhance the capacity of III.
>
> Interestingly, once we finish training TuckerBERT-144-384 and TuckerBERT-144-64, before starting inference,
> one could merge $P$ and $\mathbf{C}$ as a single matrix: $\mathbb{R} ^{12L \times 12L} \times  \mathbb{R} ^{12L \times d^2} \rightarrow \mathbb{R} ^{12L \times d^2}  $, which does not affect the model but  slightly improve inference speed. By doing so, it is rigorously equivalent to III in the inference phase. Thus, in the second version, we rename TuckerBERT-144-384 and TuckerBERT-144-64 as BERT-III-384 and   BERT-III-64 respectively.
>
> In the revised version, we have added one more section in the appendix called  "How IV  generalizes III".
>
> **Regarding MF** We did not implement II (Matrix decomposition), this is because
>
> - MF has limited potential for achieving a big compression ratio.
>   Technically, for a matrix $W \in \mathbb{R}^{m\times{n}}$, the compression ratio is $\frac{mn}{(m+n)r}$
>   For example with (sacrificing) a half rank, MF have identical parameters comparing to the raw models.
>
> - Matrix decomposition II has been exhaustedly explored by existing work (Noach \& Goldberg 2020, Mao et.al 2020), results of which were reported in this paper.
>
> **Comparision between II, III, and  IV** Experiments (see Table 10 for a full comparison, Table 4 did not show all results, we tailor the standard evaluation protocol to Noach \& Goldberg 2020 for fair comparison) show that BERT-III-384,  BERT-IV-72-384, BERT-IV-36-256, BERT-IV-36-128 outperforms  II (Noach \& Goldberg 2020) (denoted as BERT-II-245 since it uses a rank of 245 in  Noach \& Goldberg 2020).   Note that inference time of Matrix decomposition (II) with a rank $D$ is close to BERT-III or BERT-IV with the same dimension rank D; therefore, BERT-IV-36-128 is faster than BERT-II-245 (see RPS in Table 4 and inference time in Figure 8). Take BERT-IV-36-128  as an example, BERT-IV-36-128 is faster, smaller and better-performed than BERT-II-245.
>
>
>
>
>
> III (In this paper) has slight advantages over IV  in terms of performance, since  III has more parameters and therefore has better expressive power. While with the same amount of parameters, IV could have much better performance. To choose which one to use. it really depends on the trade off between  parameter scale, and performance. Generally, from  I, III, to IV,  the number of parameters decreases (or say, more parameter-efficient) while tolerancing slightly more performance drop. Note that the inference time only depends on the dimension rank in  II, III, to IV.

---

> ### Author Response · Authors · 2021-11-15
> **Replies to Reviewer 7TPf: second part**
>
> **R 4.2 why not distill the logists in GD**
> We did not distill the logits. We test to distill the logits but did not find any benefit. A similar observation could be found in Tinybert (Jiao et. al). One of the reasons is that the logits in the general distillation are specific to **the masked word prediction task** (MWP). While we finally have validated BERT in **downstream tasks** (which does not use the prediction head of **the masked language prediction task**), distilling the MWP-specific logits may not be beneficial. Instead, the hidden states are both used in MWP task and downstream tasks; therefore, we distill  the hidden states in the last layer. The reason in depth needs further investigated, while we leave it in future work.
>
> **R 4.3 Difference in the norm and scale issue**
>
> Sorry for  cross-posting with R 3.9.
>
> *Regarding approximating the raw weights (and their norm/scale)* We want to clarify that we **do not intend to approximate the raw weights (and their norm/scale)**.  Previously, we tried to add a regularizer in the training loss function to force the  reconstructed weights  using decomposition to be as close as the raw weights.
>
> $$ \mathcal{L} =  \mathcal{L}_\textrm{training} + \lambda | \mathbf{W}^{IV} - \mathbf{W}^{I} |_2  $$
>
> This does not improve the performance, but worsens the performance. The reason may be as below.  Since we cannot perfectly approximate the raw weights with a decent compression ratio, there always exists some difference between the raw weights and the reconstructed weights  using decomposition. However, even a slight difference  might lead to a big difference in output.  Plus, we also observe that the finally learned $\mathbf{W}^{IV}$ has a big difference with $\mathbf{W}^{I}$.  So we give up our efforts to approximate the raw weights.
>
> For example, we found that even randomly initializing the BERT-III/BERT-IV does not harm the final performance, compared to initialization by decomposition from a given model. This also shows that approximating raw weights may not be beneficial. Instead, we use knowledge distillation to *simulate the input-output mapping* from the original model and *tolerate the difference between raw weights and reconstructed weights*. We believe that the former makes more sense than the latter.  Empirical results show the former performs well.
>
> *Norm difference in Tucker Decomposition*  Decomposition is to approximate a  three-order tensor with three factor tensors and a core tensor.
>
> $$ \mathbf{W} _i =  (\mathbf{C}  P_i) \times _2 U \times _3 V$$
>
> Each matrix has its specific factor vector $P_i$,  which does not have any constraints.   The flexibility of norms in $P_i$ could compensate for the norm difference for the original matrices.
>
> *The norm for these weights* In case the reviewer is curious, we show it in  Appendix F.2.
>
>
>
> **R 4.4 It is furthermore not clear that the improvements are coming from the proposed approach as opposed to something else, since an ablation study across the different decomposition methods is missing.**
>
> In the revised version, we have compared II, III, to IV, as replied R 4.1. See the ablation results in the third paragraph in 6.2.

---

### Official Review · Reviewer_rMZS · 2021-10-25

**Correctness:** 3
**Technical Novelty And Significance:** 3
**Empirical Novelty And Significance:** 2
**Recommendation:** 6
**Confidence:** 5

**Main Review:**

Strengths:

1. The framework of decomposability and tensor decomposition allows this paper to encompass and explain the benefit of multiple previous work using a single viewpoint
2. The results of compression are strong. Possibility of 50x compression, albeit at 1.5-2% accuracy loss opens a lot of possibilities for on device deployment


Weakness:

1. The use of tensor decomposition for compressing neural networks has been explored extensively for CNNs, RNNs and Embeddings. The use here to compress transformers is a natural extension of the idea
2. Decomposability and low-rank nature of FFN and MHA layers has been discussed previously in the literature. The authors themselves refer to these prior works
3. Cordonnier et al 2021, further discusses the use of tucker decomposition to express MHA layers, albeit in a slightly different context.

In order to improve the paper, I recommend providing more insights into the workings of the method and the bias that the fixed structure like tucker decomposition can lead to. I also recommend exploring the systems impact of running training and inference using tucker decomposed layers and why say 50x reduction in parameter count, does not lead to a 50x improvement in RPS (Table 4). Further, I would encourage the authors to explore and understand why finetuning with KD in Table 6 leads to such large accuracy drop. GD+TD should lead to better accuracy, but improving accuracy by 40% is an interesting data point.

Questions:

1. In section 4.2, the authors say "During the inference phase, the terns that do not involve batch size b or seq length n could be calculated in an offline way...". Could you expand on what you were referring to?
2. Table 4 should have comparisons to sparsified BERT, esp for data points with 2-3x parameter compression. Both structured sparsity and random sparsity could achieve said compression. Eg - https://arxiv.org/abs/2109.04838
3. Tucker decomposition of matrices across layers forces common parameters between the matrices. A possible way this methodology could go wrong is if these matrices have different scales. Is the norm value across matrices similar for different layers? Have you looked at the problem from this point of view?

**Summary Of The Paper:**

This paper proposes to use tensor decomposition to compress the multi-head attention (MHA) and FFN layers in transformer architecture.

Premise:

Drawing from previously published research and their own experiments using PCA, the paper shows that the MHA and FFN layers are over-parameterized and exist in a lower dimensional subspace. Further, the authors talk about where these redundancy might come from and point to the decomposability of FFN and MHA layers and how that can lead to different parts learning similar behavior. Using PCA the authors show that these layers exhibit both inter-matrix and intra-matrix redundancy. Inter-matrix redundancy calls for low dimensional representation of each matrix in the layer, while intra-matrix redundancy implies possibility of parameter sharing between matrices.

Proposed Solution:

Based on this observation, the authors discuss the merit of various methods for decomposition of MHA and FFN matrices. Specifically, they discuss matrix decomposition, tensor train decomposition and tucker decomposition. The last technique will have the largest impact on compression and also has the most marginal cost of adding a new layer.

Results:

1. Impressive accuracy/parameter pareto frontier. Can compress BERT-base by ~50x with 1.5 to 2% loss in accuracy (Table 4)
2. Ablations on the need for 2 stage knowledge distillation - stage 1 to ensure attention maps and output of last layer are similar and stage two for task distillation


**Summary Of The Review:**

Overall, I think this paper is an incremental improvement to previous state-of-the-art. Tucker decomposition has been used extensively in NN to compress RNNs, CNNs and Embeddings. Thus use of tucker decomposition and its ability to compress BERT MHA and FFN layers is incremental improvement over the previous results, especially given the fact that prior work has also shown that MHA and FFN layers can be decomposed in a low rank structure and talked about the redundancy in the parameters in those layers.  However, the results of the paper are interesting from an engineering point of view.

---

> ### Author Response · Authors · 2021-11-15
> **Replies to Reviewer rMZS**
>
> **R 3.1 Difference with these works using Tensor decomposition for compressing CNNs, RNNs and Embeddings**
>
> - Compressing pre-training models is a new scenario.  We believe that exploring tensor decomposition in pre-trained language models is non-trivial.  In the pre-trained language models, we could test tensor decomposition in very deep and wide networks like GPT 3 (96 layers and a hidden dimension of 12288), while this work is a preliminary  first step. Also, the two-stage training (pre-training and fine-tuning) paradigm has a  task adaption problem; for example, the complexity of masked word prediction task may be bigger than that of downstream tasks  e.g., classification.  The two-stage training paradigm  may bring some novel mechanisms during compression.
> - Existing works (Ma et al. 2019, Liu et al. 2021,   Gao et al., 2020, Khrulkov et.al. 2019, Hrinchuk et.al. 2020, Panahi et.al.) which use tensor decomposition for compressing neural networks do not have the potential for acceleration. The bottleneck in speed limits the application of tensor decomposition in real-world applications: compressing models but consuming longer inference time seems to be useful in very rare scenarios. We argue that it is nontrivial to compress neural networks using  tensor decomposition with acceleration effects, as this work did.
> - Work mechanisms for compression are totally different, previous works  compress  each weight matrix ($W^Q, W^K, W^V, W^O, W^{in},W^{out}$ in each layer) individually using matrix/tensor decomposition. They are making use of local redundancy inside each matrix.   While in big models (PLMs), we believe that making  use of cross-matrix/cross-layer redundancy is also, sometimes more,  beneficial.  We believe  that using tensor decomposition  for cross-matrix/cross-layer redundancy  is a significant difference.
>
>
>
> We have added the above discussions in Appendix E1 in the revised version.
>
>
>
> **R 3.2 Low-rank nature of FFN and MHA layers**
> Almost all compression work in PLMs is  claimed to  be either implicitly or explicitly based on the  low-rank nature of PLMs. This is the motivation for this paper, which was not considered as one of our contributions. In the revised version, we have shortened the length of the motivation section to save more space to address the readers/reviewers' concerns.
>
> In the motivation section, we additionally explored cross-matrix/cross-layer redundancy, we believe this is relatively novel.
>
>
>
> **R 3.3 Difference with Cordonnier et al 2021**
>
> Cordonnier et al 2021 is quite impressive and inspirable. It does inspire this paper, however, we want to highlight the difference with Cordonnier et al 2021.
>
> - **Motivation** The motivation is different, Cordonnier et al 2021 found redundancy in different heads.  We make use of redundancy  of both intra-matrix redundancy and inter-matrix redundancy (including cross-layer redundancy )
>
> - **Method** The architecture in Cordonnier et al 2021 is slightly different with Transformer (or BERT) since it redesigns the self-attention network with collaborated attention. While our architecture is nearly the same as the raw model, we simulate each matrix multiplication of BERT with a product of many smaller matrices.
>
> - **Goal** Our work aims to extremely compress neural networks, which cannot be achieved by Cordonnier et al 2021.  Cordonnier et al 2021 is to explore the possibility to share key/query projection in SANs. Note that SAN only has 1/3 parameters in Transformer, this proportion even becomes smaller when considering the embedding layer. By compressing SANs with $\frac{1}{3}$, its overall compression ratio is limited.
>
> - **Potential for efficiency** Cordonnier et al 2021 is faster than the raw model only if $D_k$ is small. The smallest setting with  $D_k = 64$ has **20%** FLOPs reduction. This shows its potential for acceleration is limited.  A similar setting with $D_k = 64$ in this paper, it has a nearly **80%** reduction in terms of FLOPs.
>
> - **Potential for compression ratio** The smallest model for BERT-base (110M) in Cordonnier et al 2021 has **96.6M** when $D_k = 128$; its best compression ratio is 1.14. While in our model, the smallest model has a much bigger compression ratio, while being faster and performing better.
>
>
> - **Potential for effectiveness** The  model   in Cordonnier et al 2021  (**96.6M** parameters) achieves  **93.5%**  performance (77.6/83.0) with BERT base, while our smallest model with **25M** parameters (plus parameters in the embedding layer) achieves **97.7%**  performance (80.8/82.7) of BERT-base.  A little difference is that we test our model on the test dataset through GLUE online benchmark while Cordonnier et al 2021 test their model on offline dev dataset through average results for three runs, so we use the relative performance.
>
>
> We have added the above in Appendix E2 in the revised version.

---

> ### Author Response · Authors · 2021-11-15
> **Replies to Reviewer rMZS: second part**
>
>
> **R 3.4 Providing more insights into the workings of the method and the bias that the fixed structure like tucker decomposition can lead to.**
>
> We have one more section in the appendix to discuss the learned model based on tucker decomposition.
>
>
>
> **R 3.5 Why say 50x reduction in parameter count, does not lead to a 50x improvement in RPS**
>
> The main reason is that, when  considering parameters in the  embedding layer, the compression  ratio  is $5\times$. So the ideal upper bound of RPS smaller than  $50\times$. Note that this should consider the difference in FLOPs, we have $5 \times$ smaller FLOPs (4.3B vs. 22.5B).  The improvement in RPS is quite specific to hardware, the FLOPs and parameter reductions could provide an upper bound for RPS, while to which degree it approximates the ideal upper bound depends on the hardware and the engineering optimization, which we are not good at.
>
> We would like to highlight that extremely compressing a bigger model may enable storing a model in a single (or less) GPU server, which largely reduces the cost of network communication.  This shows that our model is potential for bigger models.
>
>
>
> **R 3.6 Results involved KD Table 6**
>
> We made a typo in the first version, which misspell **w/o** with **w/t**. Sorry to cause your confusion.
>
>
>
> **R 3.7 Offline calculation**
>
> Such calculation is not batch-specific. We could  calculate once before all inference operations and maintain the calculated matrices instead of online calculation during each inference operation.
>
> Let us explain it in a concrete example. We have two choices to calculate $ \mathbf{X} U (P_i \mathbf{C} ) V$ . Note $U (P_i \mathbf{C} ) V$ are the stored parameters ($U\in \mathbb{R}^{D\times d}, P \in \mathbb{R}^{12L \times l}, \mathbf{C} \in \mathbb{R}^{ld^2}  , V \in \mathbb{R}^{d \times D}$), while $\mathbf{X}$ is the real-time input. Note a slice of $P$ is  called $P_i \in \mathbb{R}^{1\times l}$
>
> |         | all parameters stored                                        | parameters involved in this calculation                      | # paras             | operation sequences                                          |
> | ------- | ------------------------------------------------------------ | ------------------------------------------------------------ | ------------------- | ------------------------------------------------------------ |
> | online  | $\{ U ,P, \mathbf{C}, V \}$                                  | $\{ U ,P_i, \mathbf{C}, V \}$                                | $ld^2 + 12Ll + 2Dd$ | $ \begin{cases}   h_1 = \mathbf{X} U \\\\ {\color{red} h_2 = P_i \mathbf{C} } \\\\ h_3 = h_1 h_2 \\\\ Y = h_3 V \\\\ \end{cases}$ |
> | offline | $\{ U ,\hat{\mathbf{C}}, V \}$ and $\hat{\mathbf{C}} = P \mathbf{C}$ | $\{ U ,\hat{\mathbf{C}}_i, V \}$ and $\hat{\mathbf{C}}_i = P _i\mathbf{C}$ | $12Ld^2 + 2Dd$      | $ \begin{cases}   h_1 = \mathbf{X} U \\\\ h_2 = h_1 \mathbf{C}_i \\\\ Y = h_2 V \\\\ \end{cases} $ |
>
> Compared to the online setting, the offline setting store more parameters $12Ld^2 > (ld^2 + 12Ll)$, but has fewer operation steps (3 vs. 4), since **$ P _i\mathbf{C}$  is calculated and stored in an offline way**.
>
>
>
> **R 3.8 Structured sparsity and random sparsity**
>
> We discuss Lagunas et.al. in our related work and also discuss it in the experimental results.
>
> We cannot add them to Table 4 for experimental results because they reported results in the offline dev dataset while we reported the test dataset through the online benchmark. Instead, in the revised version, we have reported our models in dev set compare with Lagunas et.al., see Table 11.

---

> ### Author Response · Authors · 2021-11-15
> **Replies to Reviewer rMZS: third part**
>
> **R 3.9 scale (norms) between matrices in SAN and FFN**
>
>
> *Regarding approximating the raw weights (and their norm/scale)* We want to clarify that we **do not intend to approximate the raw weights (and their norm/scale)**.  Previously, we tried to add a regularizer in the training loss function to force the  reconstructed weights  using decomposition to be as close as the raw weights.
>
> $$ \mathcal{L} =  \mathcal{L}_\textrm{training} + \lambda | \mathbf{W}^{IV} - \mathbf{W}^{I} |_2  $$
>
> This does not improve the performance, but worsens the performance. The reason may be as below.  Since we cannot perfectly approximate the raw weights with a decent compression ratio, there always exists some difference between the raw weights and the reconstructed weights  using decomposition. However, even a slight difference  might lead to a big difference in output.  Plus, we also observe that the finally learned $\mathbf{W}^{IV}$ has a big difference with $\mathbf{W}^{I}$.  So we give up our efforts to approximate the raw weights.
>
> For example, we found that even randomly initializing the BERT-III/BERT-IV does not harm the final performance, compared to initialization by decomposition from a given model. This also shows that approximating raw weights may not be beneficial. Instead, we use knowledge distillation to *simulate the input-output mapping* from the original model and *tolerate the difference between raw weights and reconstructed weights*. We believe that the former makes more sense than the latter.  Empirical results show the former performs well.
>
> *Norm difference in Tucker Decomposition*  Decomposition is to approximate a  three-order tensor with three factor tensors and a core tensor.
>
> $$ \mathbf{W} _i =  (\mathbf{C}  P_i) \times _2 U \times _3 V$$
>
> Each matrix has its specific factor vector $P_i$,  which does not have any constraints.   The flexibility of norms in $P_i$ could compensate for the norm difference for the original matrices.
>
> *The norm for these weights* In case the reviewer is curious, we show it in  Appendix F.2.

---

### Official Review · Reviewer_MtV7 · 2021-11-03

**Correctness:** 2
**Technical Novelty And Significance:** 3
**Empirical Novelty And Significance:** 1
**Recommendation:** 5
**Confidence:** 4

**Main Review:**

Reasons for score:

I think the idea proposed in the paper is novel, but some design choices can be further elaborated and there should be more experiments on larger models and more ablation studies. Detailed comments:

Strengths:

- Applying tensor decomposition across layers to utilize the similarity between layers is novel. This is a valuable contribution to the model compression community.
- The paper analyzes the optimal way to perform matrix multiplication given the compression method proposed in the paper.

Weaknesses:

- The paper proposed multiple potential ways of compressing weight matrices (matrix decomposition and tensor train decomposition) as some alternatives to the proposed Tucker decomposition. However, the author didn't compare with tensor train decomposition due to time constraints. I believe the paper will be more solid by adding ablation studies on tensor train decomposition.
- The paper only performs experiments on BERT-base and TinyBERT models, but I believe that the compression method proposed in the paper should be more demanded by larger models.
- Some design choices in the paper seem arbitrary. For example, why do you jointly compress all the layers, including all the FFN weights and attention weights? I can understand the similarity of the attention query weight vectors across the layers, but I can’t understand why all the weights are merged. In addition, the reason for splitting each FFN weight matrix into 4 seems to make it possible to combine it with attention weights.

Other comments:

- How does this method compare with previous works in terms of training/inference latency?

**Summary Of The Paper:**

This paper proposes to use tensor decomposition to jointly compress the model weights in all attention and FFN layers of a Transformer model, which reaches similar performance as the original BERT model while marking the model much smaller.

**Summary Of The Review:**

The paper would be better with more experiments on larger models and ablation studies. Also, the presentation and the rationale behind the idea are not clear to me.

---

> ### Author Response · Authors · 2021-11-15
> **On tensor train decomposition (as III in this work)**
>
> **R 2.1 On tensor train decomposition**
>
> Thanks for your interest in tensor train decomposition, denoted as III. We understand the concern from the reviewer about tensor train decomposition as experimental comparisons.
>
>
> Actually, **III (tensor train decomposition) is a special case of IV (tucker decomposition) in this paper**. More interestingly, we have implicitly implemented  III in the experiments; this was even not realized by ourselves when submitting this paper. In the revised paper, we will not highlight only  Tucker decomposition; instead, we exhaustively implement and investigate III  and IV.
>
> *why III (tensor train decomposition) is a special case of IV (tucker decomposition)* Note that we are not stating the correspondence between tensor train decomposition and tucker decomposition in the general case, but a three-order tensor train decomposition is a special case of tucker decomposition.
>
> $
> \begin{cases}
> \mathbf{W_i}^{III} = U \mathbf{\Sigma_i} V \\
> \mathbf{W_i}^{IV} = U (P _i \mathbf{C})  V  \\
> \end{cases}
> $
>
> Or in the other format:
> $
> \begin{cases}
> \mathbf{W}^{III} =  \mathbf{\Sigma}  \times _2 U \times _3 V  \\
>  \mathbf{W}^{IV} = (P  \mathbf{C}) \times _2 U \times _3 V  \\
>  \end{cases}
> $
>
> $\times _1$ and $\times _3$ are the mode-$1$ and mode-$3$ multiplication.
>
> The only difference between   $\Sigma \in  \mathbb{R}^ { 12L \times  d\times d}$ and $(P  \mathbf{C}) \in  \mathbb{R}^{ 12L \times  d\times d}$. In the latter,  $P  \in  \mathbb{R}^{ 12L \times l}$ and $\mathbf{C} \in \mathbb{R}^{ l \times  d\times d} $. In a special case of tucker decomposition when (1) $12L= l$ and (2) $P = I_{12L}$, it is equivalent to the 3-order tensor train decomposition (III). $ I _{12L}$ is an (square) identity  matrix with a size of $12L \times 12L$, in which the diagonal element equals to one and 0 elsewhere.
>
> In our experiments, there existed two settings previously called TuckerBERT-144-384 and TuckerBERT-144-64, which set  $12L= l = 144$ that satisfied (1). The rigorous difference with   TuckerBERT-144-384/TuckerBERT-144-64 and III is that we do not force $P$ to be equal to a identity matrix, instead $P$ is flexible to be trained. We argue that adding such a  flexible $P$ could enable crossing layer sharing and enhance the capacity of III.
>
> Interestingly, once we finish training TuckerBERT-144-384 and TuckerBERT-144-64, before starting inference,
> one could merge $P$ and $\mathbf{C}$ as a single matrix: $\mathbb{R} ^{12L \times 12L} \times  \mathbb{R} ^{12L \times d^2} \rightarrow \mathbb{R} ^{12L \times d^2}  $, which does not affect the model but  slightly improve inference speed. By doing so, it is rigorously equivalent to III in the inference phase. Thus, in the second version, we rename TuckerBERT-144-384 and TuckerBERT-144-64 as BERT-III-384 and   BERT-III-64 respectively.
>
> In the revised version, we have added one more section in the appendix called  "How IV  generalizes III".

---

> ### Author Response · Authors · 2021-11-15
> **Potential for larger models**
>
> **R 2.2 Why not a larger model?**
>
> Thanks for your suggestions, we also believe that  our models could have a better potential for larger models.  However, we do not have enough computing resources to test it in large models.
>
> In the revised version, we have added one section in the appendix to show the potential of compression in larger models.  In the section, we compare various compression ratios of II, III,  and IV, see the parameters and their compression ratios (this additionally considers parameters in the embedding layer) as below:
>
>
>
> | model        | Paras  | $L$  | $D$   | GPT-II-2048                     | GPT-III-2048                    | GPT-IV-12-768                    | GPT-IV-12-2048                   | GPT-IV-144-2048                  |
> | ------------ | ------ | ---- | ----- | ------------------------------- | ------------------------------- | -------------------------------- | -------------------------------- | -------------------------------- |
> | GPT-3 Small  | 125M   | 12   | 768   | 493.1M (0.3 $\times\downarrow$) | 647.2M (0.2 $\times\downarrow$) | 48.3M (2.6 $\times\downarrow$)   | 93.5M (1.3 $\times\downarrow$)   | 647.2M (0.2 $\times\downarrow$)  |
> | GPT-3 Medium | 350M   | 24   | 1024  | 1.3B (0.3 $\times\downarrow$)   | 1.3B (0.3 $\times\downarrow$)   | 56.7M (6.2 $\times\downarrow$)   | 102.5M (3.4 $\times\downarrow$)  | 656.2M (0.5 $\times\downarrow$)  |
> | GPT-3 Large  | 760M   | 24   | 1536  | 1.9B (0.4 $\times\downarrow$)   | 1.3B (0.6 $\times\downarrow$)   | 90.0M (8.4 $\times\downarrow$)   | 137.1M (5.5 $\times\downarrow$)  | 690.8M (1.1 $\times\downarrow$)  |
> | GPT-3 XL     | 1.3B   | 24   | 2048  | 2.5B (0.5 $\times\downarrow$)   | 1.3B (1.0 $\times\downarrow$)   | 102.3M (12.7 $\times\downarrow$) | 150.8M (8.6 $\times\downarrow$)  | 704.5M (1.8 $\times\downarrow$)  |
> | GPT-3 2.7B   | 2.7B   | 32   | 2560  | 4.2B (0.6 $\times\downarrow$)   | 1.8B (1.5 $\times\downarrow$)   | 194.4M (13.9 $\times\downarrow$) | 244.2M (11.1 $\times\downarrow$) | 797.9M (3.4 $\times\downarrow$)  |
> | GPT-3 6.7B   | 6.7B   | 32   | 4096  | 6.7B (1.0 $\times\downarrow$)   | 1.9B (3.6 $\times\downarrow$)   | 270.9M (24.7 $\times\downarrow$) | 324.7M (20.6 $\times\downarrow$) | 878.4M (7.6 $\times\downarrow$)  |
> | GPT-3 13B    | 13.0B  | 40   | 5140  | 10.4B (1.2 $\times\downarrow$)  | 2.4B (5.5 $\times\downarrow$)   | 333.6M (39.0 $\times\downarrow$) | 390.0M (33.3 $\times\downarrow$) | 943.7M (13.8 $\times\downarrow$) |
> | GPT-3 175B   | 175.0B | 96   | 12288 | 59.0B (3.0 $\times\downarrow$)  | 5.9B (29.5 $\times\downarrow$)  | 1.1B (162.1 $\times\downarrow$)  | 1.2B (151.6 $\times\downarrow$)  | 1.7B (102.4 $\times\downarrow$)  |
>
>
>
> It shows that III and IV could achieve much bigger compression ratios than II when using the same rank for hidden dimension (i.e., 2048). We claim that the proposed model has a better potential to compress bigger models.

---

> ### Author Response · Authors · 2021-11-15
> **Replies to the Reviewer MtV7**
>
> **R 2.3 why compress SAN and FFN together**
>
> If we separately compress SANs and FFNs, we would have two  matrix banks: one for SANs and one FFNs: $P^{(FFN)}  \mathbf{C}^{FFN} $ and $P^{SAN}  \mathbf{C}^{SAN} $. Each weight matrix in FFN(or SAN) is specific to a matrix bank $\mathbf{C}^{FFN}$ for FFN (or $\mathbf{C}^{SAN}$ for SAN) and its weight  vector over the bank $P^{FFN} _i$  ($P^{SAN} _i$). Note that the two matrix banks  ($\mathbf{C}^{SAN}$ and $\mathbf{C}^{FFN}$)have the most parameters since it is three-order tensor while others ($U,V,P$) are matrices.
>
> Note that  matrices in two matrix banks have the same shape, one could **merge (share) the two matrix banks (a $m$-size $d\times d$  matrix bank and a $n$-size  $m$-size $d\times d$ matrix bank) to get a single bigger ($(m+n)$-size) matrix bank**, this could boost the expressive power for both FFNs and SANs due to the bigger matrix bank (denoted as $ [  \mathbf{C}^{FFN}  ;  \mathbf{C}^{SAN} ]$).
>
>
>
> The unmerged one is a special case of the merged one. Let us define a new $P'$,  each element in which is defined as below:
> $$
> P_i' =\begin{cases}
> \big[ P_i; [ \overbrace{0, 0, \cdots 0}^{n}] \big] \;  \textrm{For SANs}\\\\
> \big[  [\overbrace {0, 0, \cdots 0}^{m}]; P_i \big]\;   \textrm{For FFNs}\\\\
> \end{cases}
> $$
> $P'_i$ is the new weight vector over the shared $(m+n)$-size matrix bank.
>
> Without losing any generality, we could relieve the zero-constrains in  $P'$ to get a general form that each element in  $P'$ is not forced to be zero. This could make FFNs and SANs share more matrix bases and therefore get better capacity.
>
> The  benefit could be empirically evidenced in Table 7: solely compressing SANs (without compressing FFNs) underperforms both compressing FFNs and SANs; although the former has much more parameters.
> Since in the shared setting, the shared matrix bank could compensate the counterpart of SANs.
>
> Another naïve motivation is to design a unified protocol for SANs and FFNs.
>
> **R 2.4 Compare with previous works in terms of training/inference latency**
>
> We have added the training/inference latency in Table 4 and RPS (request per second) in Figure 6, for the most related work (i.e., matrix decomposition based  compression Noach&Goldberg 2020  and cross-layer sharing ALBERT Lan et.al. 2019). It shows ALBERT is slightly faster and has better RPS.  The RPS or inference time of Matrix decomposition (II) with a rank $D$ is close to BERT-III or BERT-IV with the same dimension rank D. However, with an identical $D$ (in such a case, the inference time is comparable), BERT-III or BERT-IV not only have much fewer parameters than matrix decomposition (II), but also perform better (see Table 10 for full comparisons)
>
>
>
> **R 2.5 Rationale behind this work**
>
> We would like to highlight the rationale in this work, The observation using PCA shows that the weight matrices in SANs and FFNs could be approximated in a low-rank way. Moreover, the low-rank structure could also exist across weight matrices in SANs and FFNs
> PCA. Also, by merging two private weight matrix banks in SANs and FFNs, one could get a single bigger  matrix bank for both SANs and FFNs, which could have better expressive power and make the compression more parameter-efficient.
>
> We have added one more section in appendix F to better discuss the rationale to compress FFN and SAN together.

---

> ### Author Response · Authors · 2021-11-29
> **Feedback at the last minute**
>
> Thank you again for contributing time to provide comments and reviews in this paper, which have significantly improved this paper.  Can we provide any further explanations for your concerns that might still exist?
>
> We really appreciate it if you could provide any further feedback.

---

### Official Review · Reviewer_fa7a · 2021-11-04

**Correctness:** 3
**Technical Novelty And Significance:** 3
**Empirical Novelty And Significance:** 3
**Recommendation:** 6
**Confidence:** 4

**Main Review:**

Large scale pre-trained language models have demonstrated their effectiveness. However the large model size makes it difficult to deploy and compressing such models have drawn a lot of interest. This paper aims to compress PLMs to extremely small size mainly from the perspective of decomposition. It introduces several decomposition methods and makes a comprehensive comparison among them from the perspective of compressing Transformer layers. The Tucker decomposition is chosen to be the final solution due to its compression ratio.

The motivation is clear and the methods are technically sound. Though the introduced decomposition methods are not new, the adaption to the Transformer layers and corresponding analysis are comprehensive. The experimental results demonstrate the effectiveness of the method. Especially, the compressed model size is really competitive.

Some weaknesses:
1. The authors do not include embedding layer and prediction layer size in experiments, while only report the Transformer encoder size. I know that this can make the size of compressed model really amazing (e.g., 1.8M) and the compression ratio amazing (e.g., 86M/12.3M=7) but is not fair as the whole model including the embedding layer are used when deploying. If the embedding layer is added, the model size will increase a lot, and the compression ratio will decrease, which make the experimental results less surprising. But this should be made clear.
2. The authors name a lot of related works, but compare only very few of them in the experiments.
3. Some other method(s) are missing in the related works. For example: [1]

Some typos:
1. Section 5.1, "...are not exactly equal to the the raw weights...", duplicate "the"?
2. Section 6.2, "...outperforms ALBERT - the latter needs...while the latter does not...", two "latter"?

reference:
[1] Xu, Jin, et al. "NAS-BERT: Task-Agnostic and Adaptive-Size BERT Compression with Neural Architecture Search."

**Summary Of The Paper:**

This paper explores extreme parameter compression for pre-trained language model, especially BERT. It introduces and compares several tensor decomposition methods and proposes to leverage Tucker decomposition as the final solution. The compressed BERT model achieves much smaller size with promising performance.

**Summary Of The Review:**

The paper presents extreme compression on pre-trained language models. Though the introduced methods are not new, the adaptation to the Transformer layers and the analysis are interesting, and the experiments are convincing. Though there exist some weaknesses, I think the paper is of good quality, if the authors could mitigate them.

---

> ### Author Response · Authors · 2021-11-15
> **We have added the fact that compressing ratios will be smaller when word embedding layer is considered.**
>
> **R 1.1 Why not consider embedding layer**
>
> Thanks for your comment on the embedding layer.
> In the revised version, we have made it clear (in the caption of Table 4) that the compression ratio will decrease when considering the embedding layer while any decomposition-related reconstruction will cost extra time.
>
> As you may notice, we explained the reasons about *why we did not compress the embedding layer* in footnote 1. Here, we provide more discussions  (also with some quantitative comparisons) about the reasons to *not compress the embedding layer*:
>
> - Compressing embedding layer will definitely increase training/inference time, since a single embedding lookup is fast enough.
> - The embedding layer does not increase the total parameters when BERT has more transformer layers in the encoder. Since the parameters in the embedding layer are constant with respect to network depth.  Let us check the parameter number when compressing main weight matrices with a half dimension rank  (when the hidden dimension is 768, its half is 384 as the rank d, meanwhile keeping layer rank unchanged) and. Here, we also consider embedding layer for a fair comparison as suggested.
>
> | model                 | Paras  | $L$  | $D$   | $V$   | BERT-IV-($ \frac{12L}{2} - \frac{D}{2}$) | compression ratio |
> | --------------------- | ------ | ---- | ----- | ----- | ---------------------------------------- | ----------------- |
> | BERT-base-uncased     | 110M   | 12   | 768   | 30522 | 35.7M                                    | 3.1               |
> | BERT-large-uncased    | 340M   | 24   | 1024  | 30522 | 75.8M                                    | 4.5               |
> | GPT-3 Small           | 125M   | 12   | 768   | 50257 | 50.7M                                    | 2.5               |
> | GPT-3 Medium          | 350M   | 24   | 1024  | 50257 | 85.8M                                    | 4.1               |
> | GPT-3 Large           | 760M   | 24   | 1536  | 50257 | 165.5M                                   | 4.6               |
> | GPT-3 XL              | 1.3B   | 24   | 2048  | 50257 | 243.0M                                   | 5.3               |
> | GPT-3 2.7B            | 2.7B   | 32   | 2560  | 50257 | 498.0M                                   | 5.4               |
> | GPT-3 6.7B            | 6.7B   | 32   | 4096  | 50257 | 1.1B                                     | 6.3               |
> | GPT-3 13B             | 13.0B  | 40   | 5140  | 50257 | 1.9B                                     | 6.8               |
> | GPT-3 175B or “GPT-3” | 175.0B | 96   | 12288 | 50257 | 22.8B                                    | 7.7               |
>
> Note that the parameters of embeddings become more negligible when PLMs have more layers and bigger hidden dimension, in which case the compression ratio will approximate an idea upper bound ($ 8 = 2 \times 2^2$, which is linear to the  deduction times of *layer rank* and quadratic to the deduction times of *dimension rank*; in practice, we could use bigger deduction in both ranks, as we did in the experiments).
>
> - The shape of an embedding layer ($VD$) is related to the size of vocabulary, which is heterogeneous to other weight matrices in the Self-attention network and Feed-forward network ($D^2$ or $4D^2$). Therefore it is incompatible with the current compression protocol. To additionally compress the embedding layer, we might have to design a totally different compression protocol for the embedding layer, which makes this work more complicated.
>
> - The embedding layer is relatively easy to compress, see tensorized embedding (Hrinchuk et.al. 2020)  and word2ket which compress embedding with a maximum 307 and  93,675 times respectively with a slight performance drop. We believe that it is trivial to additionally compress the embedding layer on the basis of this work.
>
> Thus, we leave compressing the embedding layer as future work.
>
> Hrinchuk et.al. Tensorized Embedding Layers. Findings of EMNLP. 2020. https://aclanthology.org/2020.findings-emnlp.436/
>
> Aliakbar et.al. word2ket: Space-efficient Word Embeddings inspired by Quantum Entanglement. ICLR 2019.

---

> ### Author Response · Authors · 2021-11-15
> **Replies to Reviewer fa7a**
>
> **R 1.2 More compared works**
>
> The previous version only consider baselines that are related to matrix/tensor decomposition and knowledge distillation.  We have cited the suggested paper and added a few other related papers (like quantization, pruning)  in the related work.
>
> Some papers did not report results in the test set in GLUE benchmark, or only report part of these datasets. We made a lot of effort to tailor results from other papers to a unified evaluation protocol.
>
> Note that the proposed method is orthogonal to existing compression methods like distillation, quantization, pruning, etc., we also explore the benefit of the proposed method on a distilled BERT.
>
> **R 1.3 Missing reference**
>
> We have cited the suggested paper, Xu, et.al. as well as AutoTinyBERT (Yin et.al. ACL 2021).
>
> Also, NAS could be used to explore which layer could be shared and how to share these layers, in the context of this paper.
>
> **Typos**
> Thanks for pointing these typos out.

---

### Author Response · Authors · 2021-11-26
**General response  to all reviewers.**

Dear Reviewers,

We thank all the reviewers for their insightful and valuable comments, which have substantially improved the quality of this paper.

There are only three days (including two weekend days) for the final stage of discussions. Could you please go over our responses and revision to have more possible interactions before next Monday (29th Nov.)?   -- Especially thank reviewer rMZS for the feedback.

Based on the first-round reviews, we have revised our paper in the revision to respond to your concerns and comments.

(1) We have provided some insights to compress SAN (Self-Attention Network) and FFN (Feed-Froward Network) together. One empirical reason is that sharing matrix banks of SANs and FFNs would largely improve capacity (expressiveness).

(2) We have clarified that **IV generalizes III**. Therefore, two specific previous IV settings (i.e., BERT-IV-144-384 and BERT-IV-144-64) can be considered as III settings  (i.e., BERT-III-384 and BERT-III-64). Also, II was implemented by Noach & Goldberg 2020, Mao et.al 2020. In the revised version, we have removed the emphasis on IV settings in the whole paper, and have further explained the difference between I, II, III, and IV in terms of effectiveness and efficiency in the experiments section (Sec. 6.2)

(3) We have clarified that we do not intend to approximate the raw weights of BERT, instead, we just simulate its input-output mapping function using knowledge distillation. By doing so, we could tolerance the norm (scale) difference between reconstructed and raw weights.

(4) We have explained our potential to compress larger models.

(5)  We have added more work for comparison. Especially, we also provide the results in the dev dataset in GLUE to compare with more works (in the previous version, we only provided results in the test dataset in GLUE)

(6) We have clarified that our compression ratios will be smaller when considering the embedding layer. Also, we further explain why we did not compress the embedding layer.

Best,

The authors

---

### Author Response · Authors · 2021-11-29
**Thanks for your valuable comments and reviews**

Dear reviewers,

Thank you for reading our responses. Especially thank **rMZS** and **7TPf** for increasing their recommendation scores.

We are happy to receive more feedback from you. Let us know if there exist any (even new) concerns that were not solved or relieved.

Meanwhile, we are cleaning our codes and will make them publicly available.

Best,

The authors

---

### Decision · Program_Chairs · 2022-01-20

**Decision:**

Accept (Poster)

**Comment:**

This paper reviews a number of parameter decomposition methods for BERT style contextual embedding models. The authors argue for the application of Tucker decomposition to the attention and feedforward layers of such models. Evaluation is performed for a range of models on the GLUE benchmark. Further ablation studies indicate that the distillation procedure employed is crucial for obtaining competitive results and the raw decomposition approaches are ineffective at directly approximating the original pre-trained model.

Strengths: The reviewers generally agree that the methods explored and results presented in this paper are interesting and could be of use to those deploying large embedding models. The authors review a range of possible decomposition methods and use this to motivate their approach. The resulting levels compression are high while maintaining good performance, while the ablation study clearly shows the contribution of the various steps of the training pipeline.

Weaknesses: The main weakness identified by the reviewers is the incremental nature of this work in comparison to previous works applying various decomposition and compression techniques to neural networks. They also highlight that many of the techniques discussed early in the paper are not compared in the evaluation. The authors have effectively responded to this issue by providing further comparisons and justification for their modelling choices (e.g. not compressing the embedding layers).

Overall, despite the incremental nature of this work, I believe that there are enough though provoking ideas and results presented to warrant publication. Interestingly, as the authors emphasise in their response, the ablation study highlights that this work is not really about approximating the original models weights, as all of the work appears to be being done by the distillation procedure in concert with the choice weight decomposition. In general I wonder whether this paper would be better presented as exploring a structured distillation procedure rather than weight compression.